# CaMKII nucleates an osmotic protein supercomplex to induce cellular bleb expansion

Yuki Fujii[1], Yuji Sakai[2], Kenji Matsuzawa [1] & Junichi Ikenouchi [1✉]

## Abstract

**Blebs are membrane protrusions formed when localized regions of the plasma membrane detach from the actin cortex, enabling outward expansion driven by intracellular pressure. These structures play critical roles in cell migration and proliferation. While cortical actin contraction has been proposed as the primary driver of cytoplasmic fluid influx during bleb expansion, our prior observations revealed compartmentalization of Ca²⁺ ions and specific proteins (e.g., Mena) within expanding blebs. The functional significance of these components remained unresolved. In this study, we demonstrate that elevated Ca²⁺ levels during bleb expansion induce the assembly of a protein superstructure built around the CaMKII holoenzyme, incorporating Mena and other regulatory proteins. This complex exhibits intrinsic osmotic activity, facilitating water influx and directly contributing to bleb expansion. These findings elucidate a novel mechanism underlying bleb expansion and provide new insights into the dynamic regulation of physicochemical properties of the cytoplasm.**

**Keywords** CaMKII; Membrane Blebbing; Cytoplasmic Mechanics; Osmotically Driven Deformation; Cell Migration
**Subject Categories** Cell Adhesion, Polarity & Cytoskeleton; Signal Transduction

## Introduction

Blebs are cellular membrane structures involved in cellular motility, such as the amoeboid movement of highly motile cells such as primordial germ cells (PGCs), cancer cells and immune cells (García-Arcos et al, 2024). Furthermore, a recent study revealed that cancer cells utilize blebs not only for cell migration but also for survival, as bleb formation provides resistance to anoikis (Weems et al, 2023). Bleb formation and expansion are driven by the influx of cytoplasmic fluid into membrane protrusions where the plasma membrane has detached from the underlying actin cortex. Blebs expand rapidly while maintaining a hemispherical shape, indicating that intracellular pressure strongly drives the extension of the plasma membrane. Near the point where bleb expansion ceases, reconstruction of the actin cortex occurs, and the contraction force generated by acto-myosin cortex causes the bleb to retract (Charras et al, 2005). During the retraction process, activation of the small GTPase RhoA and its downstream effectors, including actin polymerization factors and myosin activators, has been shown to play a critical role (Aoki et al, 2016). In contrast, the mechanisms underlying the process of bleb expansion remain largely unresolved.

The cytoplasm behaves as a biphasic, poroelastic material, consisting of both solid and liquid (soluble) phases (Moeendarbary et al, 2013). The cytoplasmic solid phase is composed of cytoskeletal structures and various macromolecular assemblies such as organelles and ribosomes, and the rheological properties of the cytoplasm depend particularly on the organization of the cytoplasmic actin network (Delarue et al, 2018; Ebata et al, 2023; Fujii and Ikenouchi, 2024). This biphasic nature introduces a lag in the equilibration of cytoplasmic pressure gradients. During bleb formation, a pressure difference arises between the cell body, where the contractile actin cortex maintains high pressure, and the bleb, which lacks this cortex and thus exhibits lower pressure (Charras et al, 2005; Tinevez et al, 2009). This differential drives the expulsion of the liquid phase into the bleb, facilitating its expansion. The bleb ultimately reaches its maximum volume when the pressure equilibrates and the liquid phase becomes relatively inert.

In our previous study, we reported that the constituents and physical properties of cytoplasm within expanding blebs differ from those in other cytoplasmic regions (Aoki et al, 2021; Fujii and Ikenouchi, 2024). Specifically, we showed that both Ca²⁺ ion concentration and the levels of specific soluble proteins, such as Mena, are elevated within blebs during their expansion. When fluorescent quantum dots are introduced into the cytoplasm of blebbing cells, they display comparatively greater movement within expanding blebs and their motion is drastically reduced once the bleb begins to retract (Aoki et al, 2021). Considering that the mesh size of the cytoplasmic actin network, which largely defines the physical properties of the cytoplasmic solid phase, is typically around 30–60 nm (Mitchison et al, 2008), these findings suggest that the actin meshwork is largely absent or extremely sparse within the expanded bleb. During the retraction phase, the actin meshwork is restored in the cytoplasm, coinciding with the reformation of the actin cortex (Chikina et al, 2019), and the mobility of quantum dots decreases accordingly. In addition to these changes in the solid-phase cytoskeletal components, we observed that the behavior of liquid-phase components—such as proteins and ions—

[1]Department of Biochemistry, Kyushu University Graduate School of Medical Sciences, Fukuoka 812-8582, Japan. [2]School of Science/Graduate School of Nanobioscience, Yokohama City University, Yokohama, Japan. ✉E-mail: ikenouchi.junichi.033@m.kyushu-u.ac.jp

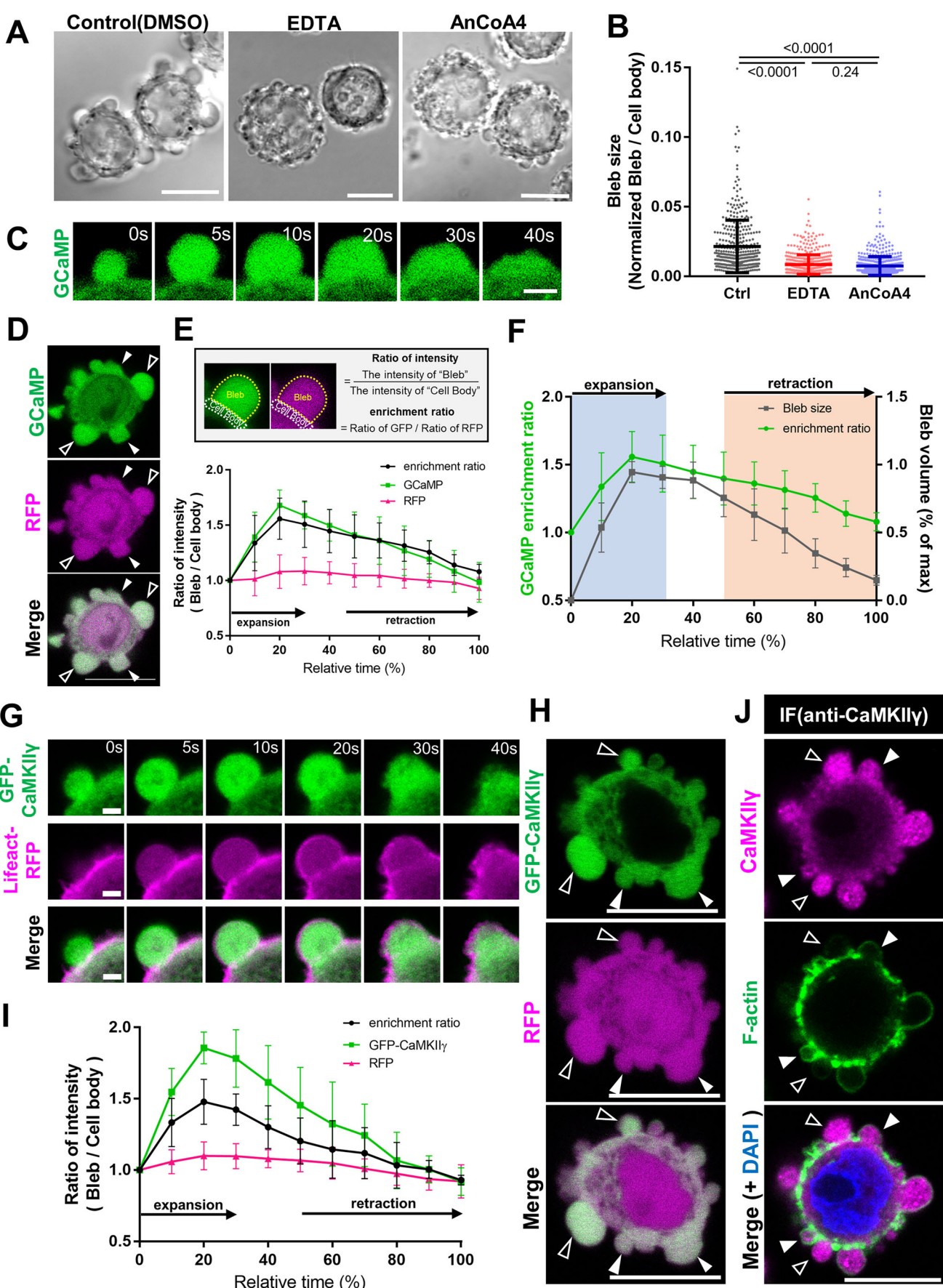

**Figure 1.  CaMKII accumulates in expanding blebs.**

(A) Phase contrast images of DLD1 WT cells treated with indicated reagents. Inhibition of extracellular Ca²⁺ influx through low-calcium condition (EDTA) or SOCE inhibitor (AnCO4) results in cells forming small blebs. EDTA is a broad chelator of divalent cations but given the similar suppression of bleb formation by SOCE inhibition, the observed inhibitory effect is predominantly attributable to Ca²⁺ depletion (Scale bar: 10 μm). (B) Distribution of bleb sizes in cells treated with various reagents. Bleb areas were quantified in 20 cells for all blebs and normalized to the cell body area. Results are shown as mean of four biological replicates ± SD and the $p$ values of one-way ANOVA followed by Tukey's post hoc test are indicated; $p$ values are $p < 0.0001$ (for Ctrl vs. EDTA), $p < 0.0001$ (for Ctrl vs. AnCoA4), and $p = 0.2415$ (for EDTA vs. AnCoA4). (C) Time-lapse images of blebs in GCaMP6s-expressing DLD1 cells. An increase in fluorescence intensity was observed within the expanding blebs. Numbers in the top-right corner indicate time (Scale bar: 2 μm). (D) GCaMP6s and RFP co-expressed in DLD1 cells. Black arrowheads indicate expanding blebs, and white arrowheads indicate retracting blebs (Scale bar: 10 μm). (E) Quantification and time course of molecular accumulation within blebs. Top: Example of quantified regions. Fluorescence intensities of each channel were measured in both the bleb and cell body regions, and the ratio was calculated. Bottom: Temporal changes in fluorescence intensity ratios and enrichment rates of GCaMP6s and RFP. Each dataset represents quantification of five blebs, and the mean ± SD is plotted for each time point. The relative time was defined such that 0% corresponds to the onset of bleb formation and 100% corresponds to the completion of bleb retraction. (F) Relative change in bleb volume plotted with the change in GCaMP6s accumulation in blebs. Each dataset represents quantification of five blebs, and the mean ± SD is plotted for each time point. (G) Time-lapse images of blebs in DLD1 cells expressing GFP-CaMKIIγ and RFP-Lifeact. GFP-CaMKIIγ accumulates within expanding blebs. Numbers in the top-right corner indicate time (Scale bar: 2 μm). (H) GFP-CaMKIIγ and RFP co-expressed in DLD1 cells. Black arrowheads indicate expanding blebs, and white arrowheads indicate retracting blebs (Scale bar: 10 μm). (I) Quantification and time course of molecular accumulation within blebs. Temporal changes in fluorescence intensity ratios of GFP-CaMKIIγ and RFP were quantified. Each dataset represents quantification of five blebs, and the mean ± SD is plotted for each time point. (J) Immunofluorescence image of endogenous CaMKIIγ in wild-type DLD1 cells during bleb formation. Black arrowheads indicate expanding blebs, and white arrowheads indicate retracting blebs (Scale bar: 10 μm). Source data are available online for this figure.

also differs markedly between the expansion and retraction phases. Specifically, proteins like Mena accumulate in expanding blebs, and Ca²⁺ levels are elevated in these regions compared to the rest of the cytoplasm, as demonstrated using GCaMP-based analyses (Aoki et al, 2021). Our interpretation is that the increased fluidity within the bleb facilitates the influx and movement of the liquid phase, thereby promoting bleb expansion.

Despite these insights, a key open question remains: what is the significance of the qualitative difference in the liquid phase of the bleb cytoplasm compared to the cell body? In particular, the precise roles of calcium and the accumulation of specific soluble proteins, such as Mena, in driving bleb expansion are not yet fully understood. In this study, we aimed to clarify how these unique features of cytoplasm expanding bleb contribute to expansion. Through visual screening, we identified CaMKII proteins, which accumulates in the expanding bleb, plays an essential role in the expanding bleb. Furthermore, we found that the elevation of Ca²⁺ levels during the expansion phase induces the formation of a supramolecular complex comprising CaMKII, Mena, and other proteins. This complex becomes excluded from the cytoplasmic actin meshwork, leading to its accumulation in expanding blebs, which lack an actin meshwork. The enrichment of this supramolecular complex generates an osmotic gradient that promotes the influx of liquid-phase cytoplasmic components, thereby driving bleb expansion.

## Results

### CaMKII accumulates inside expanding blebs and contributes to their expansion

We previously demonstrated that calcium influx from the extracellular environment via store-operated calcium entry (SOCE) is essential for bleb expansion (Aoki et al, 2021). Specifically, treating cells with EDTA, a chelator of extracellular Ca²⁺ ions, or with SOCE inhibitors significantly impaired bleb expansion (Fig. 1A,B). Bleb dynamics relative to the overall time course are largely constant, with blebs achieving maximum volume rapidly

then decreasing in volume at a steady rate and GCaMP-based analyses further revealed a specific elevation in intracellular Ca²⁺ levels within expanding blebs (Fig. 1C–F). We had also shown that intracellularly introduced quantum dots exhibited significantly greater movement within expanding blebs compared to retracting blebs or non-bleb regions. Importantly, treatment with a calcium ionophore further enhanced quantum dot mobility. These observations collectively suggest that localized elevation of Ca²⁺ in expanding blebs promoted their expansion by increasing cytoplasmic fluidity (Aoki et al, 2021). However, we could not exclude the possibility that the increase in intracellular Ca²⁺ also contributes to bleb expansion by activating myosin and thereby promoting the rise in intracellular pressure required for bleb expansion, perhaps by constricting the bleb base. Therefore, further investigation is needed to clarify the specific role of localized Ca²⁺ elevation in facilitating bleb expansion.

To address this, we aimed to identify Ca²⁺-regulated molecules contributing to bleb expansion. Using GFP-tagged Ca²⁺-associated proteins in human colon cancer DLD1 cells, we performed live-cell imaging to screen for factors that accumulate in expanding blebs. We examined the localization dynamics of several GFP-tagged cytoplasmic proteins whose activities are regulated by Ca²⁺ in wild-type DLD1 cells, including CaMKI, CaMKII, CaMKKβ, PKCα, PKCγ, and ANXA5, during bleb expansion (Fig. EV1A–C). This approach identified Ca²⁺/calmodulin (CaM)-dependent protein kinase II gamma (CaMKIIγ) as a novel component enriched in the cytoplasm of expanding blebs. Notably, CaMKIIγ accumulation was transient, peaking during expansion and decreasing during retraction (Fig. 1G–I). To exclude the possibility that the increased GFP-tagged CaMKIIγ fluorescence in expanding blebs was caused by pH-dependent or other environmental effects, we performed anti-GFP immunostaining, which confirmed that GFP-tagged CaMKIIγ indeed accumulates in expanding blebs (Appendix Fig. S1). Furthermore, immunofluorescence staining for CaMKIIγ revealed that endogenous CaMKIIγ also accumulates inside expanding blebs (Fig. 1J). The physiological functions of CaMKII is long established in phenomena such as synaptic enlargement during long-term potentiation (LTP) in neurons (Bayer and Giese, 2025; Lisman et al, 2012; Yasuda et al, 2022). However, a potential

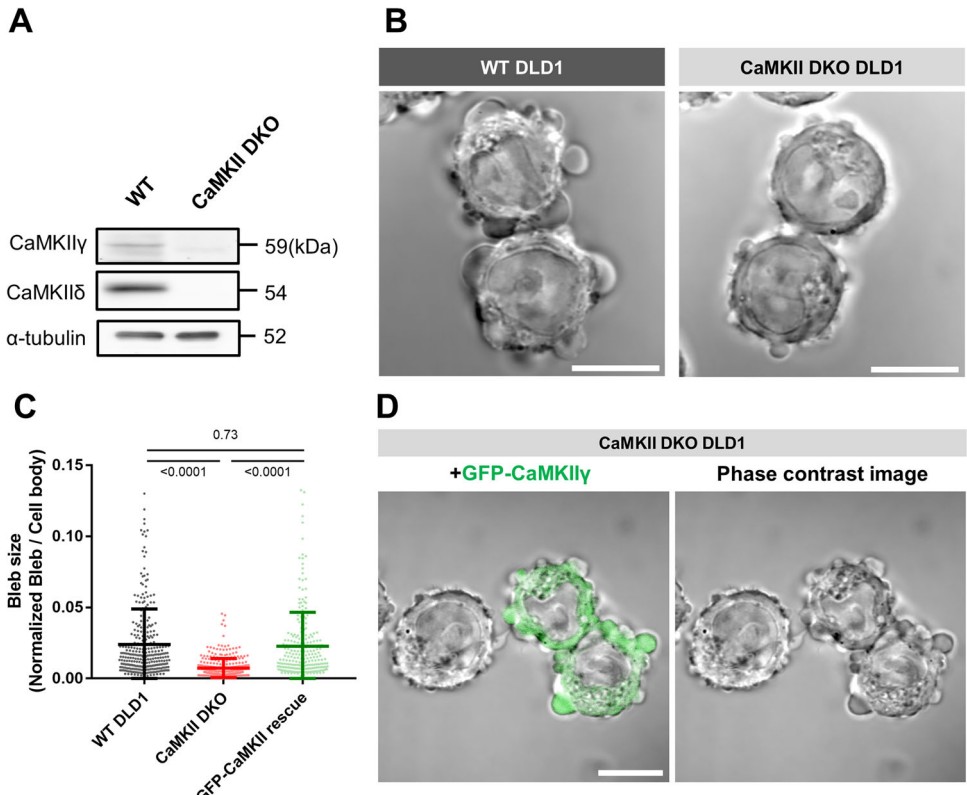

**Figure 2. Loss of CaMKII expression impairs bleb expansion.**

(A) Western blot analysis of WT DLD1 cells and CaMKII double-knockout (DKO) DLD1 cells. Expression of CaMKIIγ and CaMKIIδ was confirmed in each cell type. α-Tubulin was used as a loading control. (B) DIC images of WT cells and CaMKII DKO cells under bleb observation conditions. CaMKII DKO cells formed smaller blebs compared to WT cells (Scale bar: 10 µm). (C) Distribution of bleb sizes in WT, CaMKII DKO and CaMKII DKO cells expressing GFP-CaMKIIγ. The areas of all blebs in 20 cells were quantified. Results are shown as mean of three biological replicates ± SD and the $p$ value of one-way ANOVA followed by Tukey's post hoc test is indicated; $p$ values are $p < 0.0001$ (for WT vs. DKO), $p < 0.0001$ (for DKO vs. rescue), and $p = 0.7334$ (for WT vs. rescue). (D) Phase-contrast images of CaMKII DKO cells transfected with GFP-CaMKIIγ. In these images, cells expressing GFP-CaMKIIγ display restored bleb sizes, suggesting that the reintroduction of CaMKIIγ rescues the bleb formation defect seen in CaMKII DKO cells (Scale bar: 10 µm). Source data are available online for this figure.

role in the context of blebs had not been proposed, prompting us to probe further.

To investigate the potential role of CaMKII in bleb dynamics, we generated CaMKII-deficient DLD1 cells. DLD1 cells primarily express the γ and δ isoforms of CaMKII. Using CRISPR-Cas9, we created DLD1 cell lines lacking both isoforms (CaMKII DKO), which was confirmed by western blotting (Fig. 2A). Observation of bleb dynamics in these cells revealed that maximum bleb sizes were significantly smaller than those in wild-type (WT) cells (Fig. 2B,C). Moreover, since the expression of CRISPR-resistant CaMKIIγ in CaMKII DKO cells restored bleb size to WT levels (Fig. 2C,D), we conclude that CaMKII specifically affects bleb expansion.

## CaMKII promotes bleb expansion independent of its kinase activity

CaMKII is composed of four domains: kinase, autoinhibitory, linker, and hub domains (Fig. 3A). It forms stable hetero-dodecamers via the hub domain (Rosenberg et al, 2005). Under basal conditions, its kinase activity is suppressed by intramolecular interactions between the kinase and autoinhibitory domains (Fig. 3A) (Bhattacharyya et al, 2020). Upon binding of the $Ca^{2+}$/

CaM to the autoinhibitory domain, CaMKII adopts an activation-competent extended conformation, exposing its substrate-binding site and enabling kinase activity (Hunter and Schulman, 2005). Steric activation then induces autophosphorylation at threonine 287 (T287) within the autoinhibitory domain, rendering CaMKII constitutively active (Barria et al, 1997). Autophosphorylation at T287 has long been implicated in converting transient $Ca^{2+}$ signals into prolonged CaMKII activity; more recent evidence refines this view by assigning a role in induction/encoding rather than storage of plasticity (Bayer and Giese, 2025). Conversely, phosphorylation of T305 and T306 inhibits $Ca^{2+}$/CaM binding, leading to inactivation of CaMKII (Hanson and Schulman, 1992). Additionally, the activated holoenzyme is capable of binding other molecules. Intriguingly, interaction with GluN2B, itself a tetramer, not only stabilizes the active state of CaMKII but also promotes its liquid–liquid phase separation (LLPS) by mediating multivalent cross-linking between CaMKII dodecamers (Cai et al, 2021; Hosokawa et al, 2021).

Given these molecular characteristics of CaMKII, we first examined whether the kinase activity of CaMKII contributes to bleb expansion. We established CaMKII DKO DLD1 cell lines stably expressing GFP-tagged wild-type CaMKIIγ, an

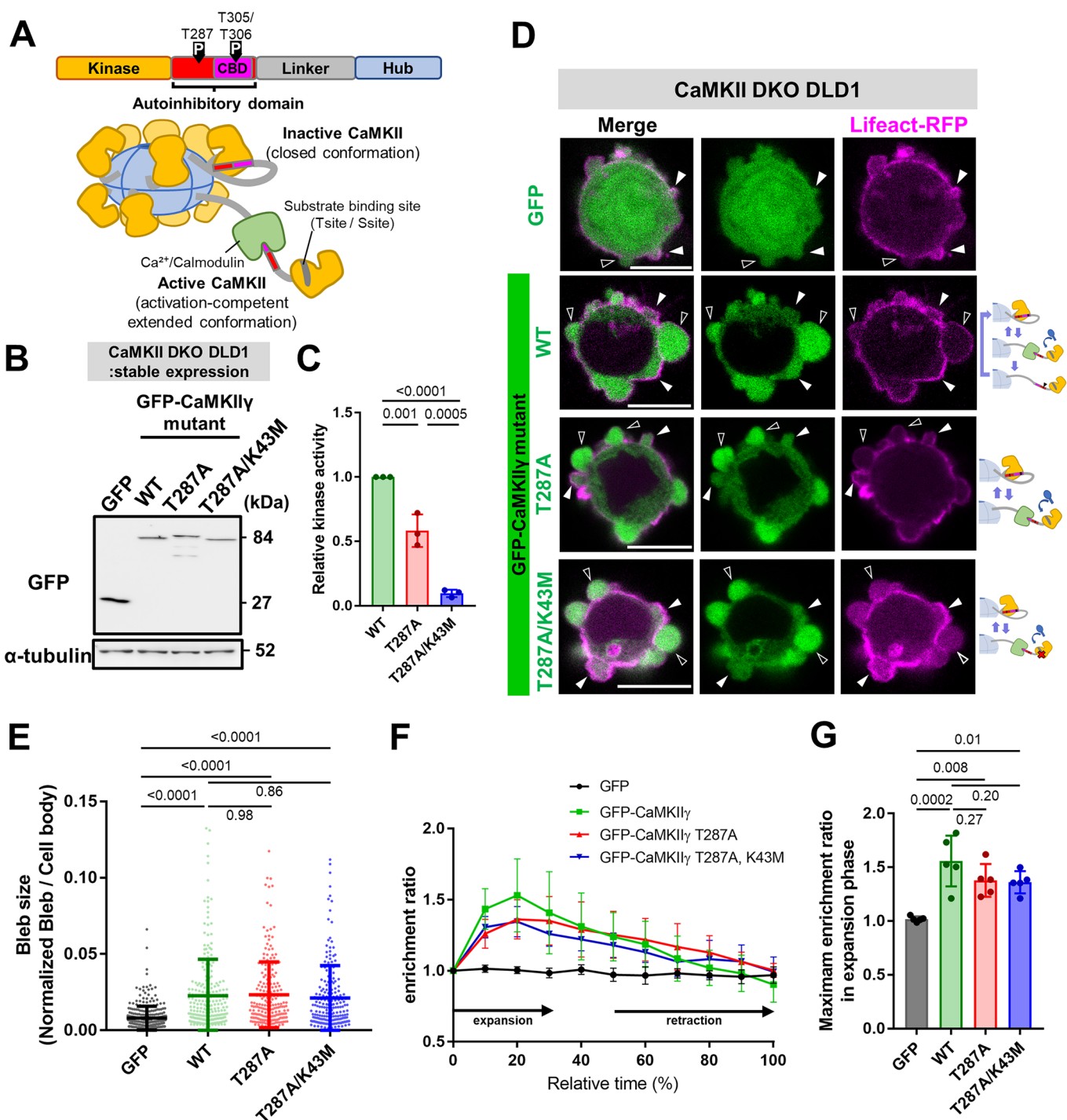

autophosphorylation-deficient mutant (T287A) (Hanson et al, 1994), and a kinase-dead mutant (T287A/K43M), in which the conserved lysine in the ATP-binding site (K43M; corresponding to K42M in CaMKIIα) (Rich and Schulman, 1998) was substituted together with the T287A mutation (Fig. 3B). Measurement of kinase activity revealed that both T287A- and T287A/K43M-expressing cells showed markedly reduced kinase activity, comparable to that of GFP-expressing control cells (Fig. 3C). The autophosphorylation-deficient T287A CaMKIIγ mutant exhibited

accumulation within expanding blebs at levels comparable to WT CaMKIIγ (Fig. 3D,F,G). Furthermore, in DKO cells expressing the T287A CaMKIIγ mutant, bleb size was restored to levels similar to those observed with WT CaMKIIγ expression (Fig. 3D,E). These results suggest that sustained kinase activity of CaMKII via autophosphorylation is not necessary for its function during bleb expansion. We therefore opted to use the T287A mutant as a control in subsequent experiments since it allowed us to exclude autophosphorylation-dependent activity change as a confounding

**Figure 3. Kinase activity of CaMKII is not required for regulating bleb expansion.**

(A) Schematic of CaMKII domain structure. (B) Western blot analysis of CaMKII DKO DLD1 cells stably expressing GFP-tagged CaMKIIγ mutants. α-Tubulin was used as a loading control. (C) Kinase activity of CaMKII DKO DLD1 cells stably expressing GFP-tagged CaMKIIγ mutants. Data represents the mean ± SD from three independent experiments. Statistical significance was assessed by one-way ANOVA followed by Tukey's post hoc test; $p$ values are $p = 0.0012$ (for WT vs. T287A), $p < 0.0001$ (for WT vs. T287A/K43M), and $p = 0.0005$ (for T287A vs. T287A/K43M). (D) CaMKII mutants tagged with GFP and Lifeact-RFP were expressed in CaMKII DKO cells. Black arrowheads indicate expanding blebs, while white arrowheads indicate retracting blebs (Scale bar: 10 μm). (E) Distribution of bleb sizes in DLD1 CaMKII DKO cells expressing various CaMKII mutants. Non-expressing cells were used as a control and labeled as DKO. Bleb areas were quantified for all blebs in 20 cells per group. Results are shown as mean of three biological replicates ± SD and the $p$ values of one-way ANOVA followed by Tukey's post hoc test is indicated; $p$ values are $p < 0.0001$ (for GFP vs. WT), $p < 0.0001$ (for GFP vs. T287A), $p < 0.0001$ (for GFP vs. T287A/K43M), $p = 0.9844$ (for WT vs. T287A), $p = 0.8553$ (for WT vs. T287A/K43M), and $p = 0.6805$ (for T287A vs. T287A/K43M). (F) Time-dependent changes in enrichment of various GFP-tagged CaMKII mutants relative to monomeric RFP within expanding blebs of CaMKII DKO cells. Each dataset represents quantification of five blebs, and the mean ± SD is plotted for each time point. (G) Maximum enrichment ratio of GFP–CaMKII mutants relative to monomeric RFP within blebs of CaMKII DKO cells. The peak ratio value was determined for each bleb, and quantification was performed for five blebs per group. Statistical significance was evaluated by one-way ANOVA followed by Tukey's post hoc test; $p$ values are $p = 0.0002$ (for GFP vs. WT), $p = 0.0079$ (for GFP vs. T287A), $p = 0.0117$ (for GFP vs. T287A/K43M), $p = 0.2688$ (for WT vs. T287A), $p = 0.1997$ (for WT vs. T287A/K43M), and $p = 0.9974$ (for T287A vs. T287A/K43M). Source data are available online for this figure.

factor in interpreting any results. The T287A/K43M mutant accumulated within expanding blebs at levels comparable to those of WT CaMKIIγ (Fig. 3D,F,G). When expressed in CaMKII DKO cells, bleb size recovered to a level in line with WT CaMKIIγ-rescued cells (Fig. 3D,E). These results suggest that CaMKII contributes to bleb expansion through a mechanism independent of its kinase activity, highlighting a novel, non-catalytic function of CaMKII in bleb expansion.

## Structural changes of CaMKII are critical for bleb expansion

Since enzymatic activity of CaMKII was dispensable for bleb expansion, we questioned whether its structural state contributes to its function regarding bleb expansion. CaMKII undergoes conformational changes upon binding to Ca²⁺/CaM. To assess whether this interaction is required for bleb expansion, we treated DLD1 cells with KN93, a compound that specifically binds to Ca²⁺/CaM and prevents its interaction with CaMKII. KN93 treatment strongly inhibited bleb expansion, similar to the effect observed under EDTA treatment (Fig. 4A,B). We next sought to determine whether conformational changes in CaMKII play a critical role in bleb expansion by visualizing the structural state of CaMKII within blebbing cells. For this purpose, we utilized the Camui-CR biosensor, which reflects CaMKII structural activation. Camui-CR consists of full-length CaMKIIα fused with Clover at the N-terminus and mRuby2 at the C-terminus (Lam et al, 2012). As illustrated in the schematic diagram in Fig. 3A when intracellular calcium levels rise and Ca²⁺/CaM binds to CaMKII, CaMKII adopts an activation-competent extended conformation (Sloutsky et al, 2020; Tsujioka et al, 2023), resulting in a decrease in FRET efficiency of Camui-CR (Lam et al, 2012) (Fig. 4C). To verify that conformational changes of CaMKII can be detected in DLD1 cells using Camui-CR, we treated adherent DLD1 cells expressing Camui-CR with different reagents and monitored changes in FRET efficiency. Upon calcium ionophore treatment, FRET efficiency decreased throughout the cells, whereas treatment with either EDTA or KN93 led to an increase in FRET efficiency (Fig. 4C). Live imaging of Camui-CR-expressing DLD1 cells revealed that Camui-CR exhibited lower FRET efficiency in expanding bleb as compared to retracting blebs or non-bleb cytoplasmic regions (Figs. 4D–G and Movie EV1). This observation indicates that CaMKII accumulates within expanding blebs in an activation-competent extended conformation.

Importantly, in cells treated with either EDTA or KN93, GFP-CaMKII failed to accumulate within expanding blebs (Fig. 5A–C), supporting the idea that an activation-competent extended conformation of CaMKII is required for its recruitment to blebs and for bleb expansion (Appendix Fig. S2A,B). To further investigate this issue, we analyzed a series of CaMKIIγ mutants with alterations in amino acid residues involved in structural changes for their ability to accumulate within blebs and their capacity to rescue the phenotype when expressed in CaMKII DKO cells (Fig. 5D,E). First, the ΔC mutant, which lacks the Hub domain necessary for CaMKII dodecamer formation, was diffusely distributed throughout the cytoplasm and did not accumulate within expanding blebs and it failed to rescue the impaired bleb expansion phenotype in CaMKII DKO cells (Fig. 5E–H). The phosphomimetic mutants that inhibit CaM binding to CaMKIIγ (T287A, T305D, T306D) were likewise unable to recapitulate WT CaMKIIγ behavior and function (Fig. 5E–H). These results are consistent with those obtained from EDTA and KN93 treatments, underscoring the critical role of Ca²⁺/CaM -induced structural change for bleb expansion.

Next, we found that the I205K T287A mutant that disrupts the hydrophobic pocket within the CaMKII kinase domain, essential for substrate binding, were also defective (Figs. 5E–H). Notably, the I205K T287A mutation does not affect the structural change induced by Ca²⁺/CaM binding but is critical for interactions with substrate proteins (Jiao et al, 2011). Previous studies have reported that substrate proteins can multimerize or form complexes with other proteins, facilitating LLPS of CaMKII dodecamers (Hosokawa et al, 2021). These findings suggest that CaMKII dodecamers not only undergo structural changes upon Ca²⁺/CaM binding but also interact with other molecules to form larger molecular complexes. This interaction likely facilitates the accumulation of CaMKII within blebs, highlighting the importance of both structural changes and molecular interactions in the role of CaMKII during bleb expansion.

## CaMKII conformational change and supramolecular assembly in expanding blebs

Based on these observations, we hypothesized that CaMKII accumulates in expanding blebs and promotes bleb expansion by forming supercomplexes with its substrates and associated proteins.

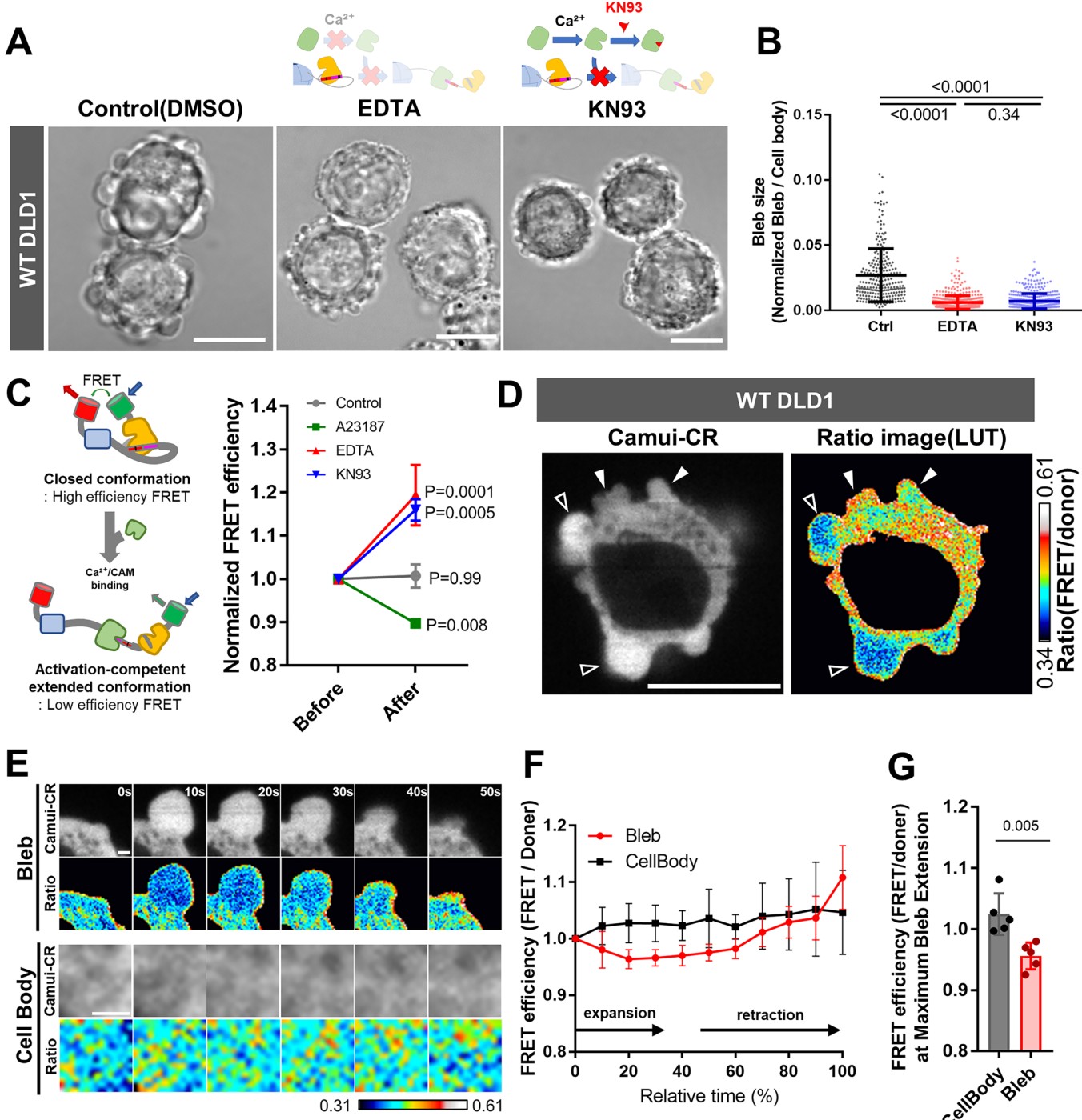

While numerous binding partners of CaMKII through the substrate-binding pocket of CaMKII, have been identified in neurons (Özden et al, 2022), the interaction partners of CaMKII in blebs are unknown. We previously identified several proteins that specifically accumulate within expanding blebs (Aoki et al, 2021). By screening these candidates in WT and CaMKII DKO cells (Fig. EV2A–C), we found that Mena, a nucleation and bundling factor for actin tetramer formation, strongly accumulated in expanding blebs in WT cells but did not in CaMKII DKO cells

(Fig. 6A–C and Movie EV2). CaMKII and Mena directly interacted and expression of the mScarlet-CaMKII T287A mutant restored GFP-Mena accumulation within expanding blebs of CaMKII DKO cells (Fig. 6D–F). By contrast, neither RFP nor the mScarlet-CaMKII I205K T287A mutant lacking substrate binding ability could restore GFP-Mena accumulation within expanding blebs (Fig. 6E,F). Extracellular Signal-Regulated Kinase 1 (ERK1, also referred to as MAPK3) was another molecule that required CaMKII to accumulate within expanding blebs (Fig. 6G–I). However, direct

**Figure 4. CaMKII promotes bleb expansion through Ca²⁺-dependent structural changes.**

(A) Phase-contrast images show WT cells treated with various reagents. Under both EDTA treatment (chelates Ca²⁺) and KN93 treatment (inhibits CaMKII conformational activation), cells display markedly smaller blebs relative to untreated controls (Scale bar: 10 µm). (B) Distribution of bleb sizes in WT cells treated with indicated agents. The areas of all blebs in 20 cells were quantified. Results are shown as mean of four biological replicates ± SD and the *p* values of one-way ANOVA followed by Tukey's post hoc test are indicated; *p* values are *p* < 0.0001 (for Ctrl vs. EDTA), *p* < 0.0001 (for Ctrl vs. AnCoA4), and *p* = 0.3383 (for EDTA vs. AnCoA4). (C) Schematic of the CaMKII FRET probe Camui-CR (left) and validation of its ability to accurately report CaMKII conformational changes in adherent, non-blebbing DLD1 cells (right). FRET efficiency was quantified before and after drug treatment in each experiment. The ratios were plotted after normalizing the post-treatment values to the pre-treatment values. Each experiment was performed three times and *p* values of two-way ANOVA followed by Tukey's post hoc test are indicated; *p* values are *p* = 0.9975 (for Control), *p* = 0.0082 (for A23187), *p* = 0.0001 (for EDTA) and *p* = 0.0005 (for KN93). (D) Fluorescence microscopy images in WT cells expressing the CaMKII FRET probe Camui-CR. The left panel shows fluorescence images of Camui-CR, while the right panel shows FRET/donor ratio images. Reduced FRET efficiency was observed in expanding blebs. (E) Time-lapse images of blebs and adjacent cell body regions in Camui-CR-expressing WT cells. Within blebs, FRET efficiency decreased during expansion but increased during retraction. In contrast, no time-dependent changes in FRET efficiency were observed in the cell body regions adjacent to the blebs. Numbers in the top-right corner indicate time (Scale bar: 1 µm). (F) Temporal changes in FRET efficiency of Camui-CR within blebs and adjacent cell body regions. Quantification was performed on five blebs per condition. Each dataset represents quantification of five blebs, and the mean ± SD is plotted for each time point. (G) FRET efficiency of Camui-CR within blebs and adjacent cell body regions at the point of maximal bleb expansion in wild-type cells. FRET efficiency was quantified for five blebs, comparing bleb and cell body regions. Statistical significance was assessed using Student's t-test; *p* value is *p* = 0.0053. Source data are available online for this figure.

interaction between CaMKII and ERK1 was not detected in our hands, suggesting that ERK1 is incorporated into a CaMKII-based supercomplex via interaction with other substrates.

Structural studies have shown that the CaMKII dodecamer adopts a disk-like structure with a diameter in the range of 15–35 nm (Myers et al, 2017). The individual CaMKII molecules then take on an activation-competent extended conformation in the presence of Ca²⁺/CaM, allowing them to each tether substrates and other protein partners, resulting in further bulk. Considering that pore sizes of the actin meshwork in the cytoplasm of the cell body typically measures 30–60 nm (Mitchison et al, 2008) and macromolecules such as ribosomes and intracellular organelles cooperate with the actin meshwork to form a porous, viscoelastic cytoplasmic solid phase with small pores, which restricts the diffusion of small molecules (Fujii and Ikenouchi, 2024; Moeendarbary et al, 2013), these CaMKII complexes are unlikely to freely diffuse through the cytoplasmic actin meshwork.

It is reported that the cytoplasmic actin meshwork is also disrupted during bleb expansion, in addition to the cortical actin network, and our previous analysis of quantum dot dynamics corroborate this (Aoki et al, 2021). Moreover, using the highly photostable Lifeact-StayGold probe to visualize actin dynamics in blebbing cells by high-speed super-resolution microscopy revealed a fine, mesh-like actin network in the cytoplasm and within retracting blebs but crucially, not in expanding blebs (Fig. EV3A,B); the presence of a cytoplasmic actin meshwork in the cytoplasm and in retracting blebs has also been documented by correlative platinum replica electron microscopy (Chikina et al, 2019). Thus, bleb expansion is marked by a transient depletion of the cytoplasmic actin meshwork, whereas retraction involves the reassembly of both the cytoplasmic actin network and the actin cortex to ultimately drive bleb retraction. In the steady state cytoplasm, the closed, compact CaMKII dodecamer can diffuse freely through the actin meshwork. Within expanding blebs, however, the actin meshwork impedes the return of CaMKII to the cell body cytoplasm, due to the formation of the Ca²⁺/CaM-dependent supercomplex. This ratchet-like molecular mechanism thus facilitates the accumulation of CaMKII in expanding blebs.

This hypothesis predicts that loosening of the cytoplasmic solid phase in the cell body would suppress local molecular condensation. As noted above, the mesh size of the cytoplasmic solid phase is defined not only by the actin cytoskeleton but also by densely

packed macromolecules such as ribosomes. Indeed, a previous study reported that treatment with the mTOR inhibitor rapamycin reduces the number of ribosomes, and consequently increases the effective mesh size to promote the diffusion of molecules with sizes on the order of several tens of nanometers (Delarue et al, 2018).

To investigate whether a reduction in ribosome abundance affects the enrichment of CaMKII within blebs, DLD1 cells were treated with rapamycin. As an initial validation, we confirmed that rapamycin treatment effectively inhibited mTORC1 signaling, as evidenced by a marked reduction in phospho-p70S6K levels (Fig. EV4A,B). Consistent with this inhibition, Western blot analysis showed that the level of ribosomal protein S3 (RPS3), a structural component of ribosomes, was also reduced in rapamycin-treated DLD1 cells compared with DMSO-treated controls (Fig. EV4A,C). Interestingly, rapamycin treatment abolished the accumulation of GFP-CaMKII within expanding blebs and led to the formation of smaller blebs compared with control cells (Fig. EV4D–G).

These observations indicate that CaMKII forms supramolecular complexes through oligomerization and multivalent interactions with its substrates within expanding blebs, and that these assemblies cannot readily diffuse back into the cell body because the cytoplasmic solid phase there has a much smaller mesh size, defined by the actin network and the abundant macromolecules such as ribosomes. Consequently, the assemblies remain confined within the bleb, leading to their robust local accumulation.

## Osmotic contribution of CaMKII-based complexes to bleb growth (CODE: CaMKII-based osmotically-driven deformation)

Then, how does the exclusion of the CaMKII supercomplexes contribute to bleb expansion? We hypothesize that these agglomerates establish a significant and abrupt change in protein concentration between the bleb and cell body cytoplasm that then generate osmotic activity to promote the influx of liquid-phase fluid to protrude blebs. Although small molecules such as ions and low-molecular-weight compounds are well-established as osmolytes, the potential for specific proteins to act as osmotic agents within cells has not been extensively explored. Proteins, unlike ions or small molecules that are present at millimolar concentrations, exist in significantly lower molecular counts. Additionally, their free

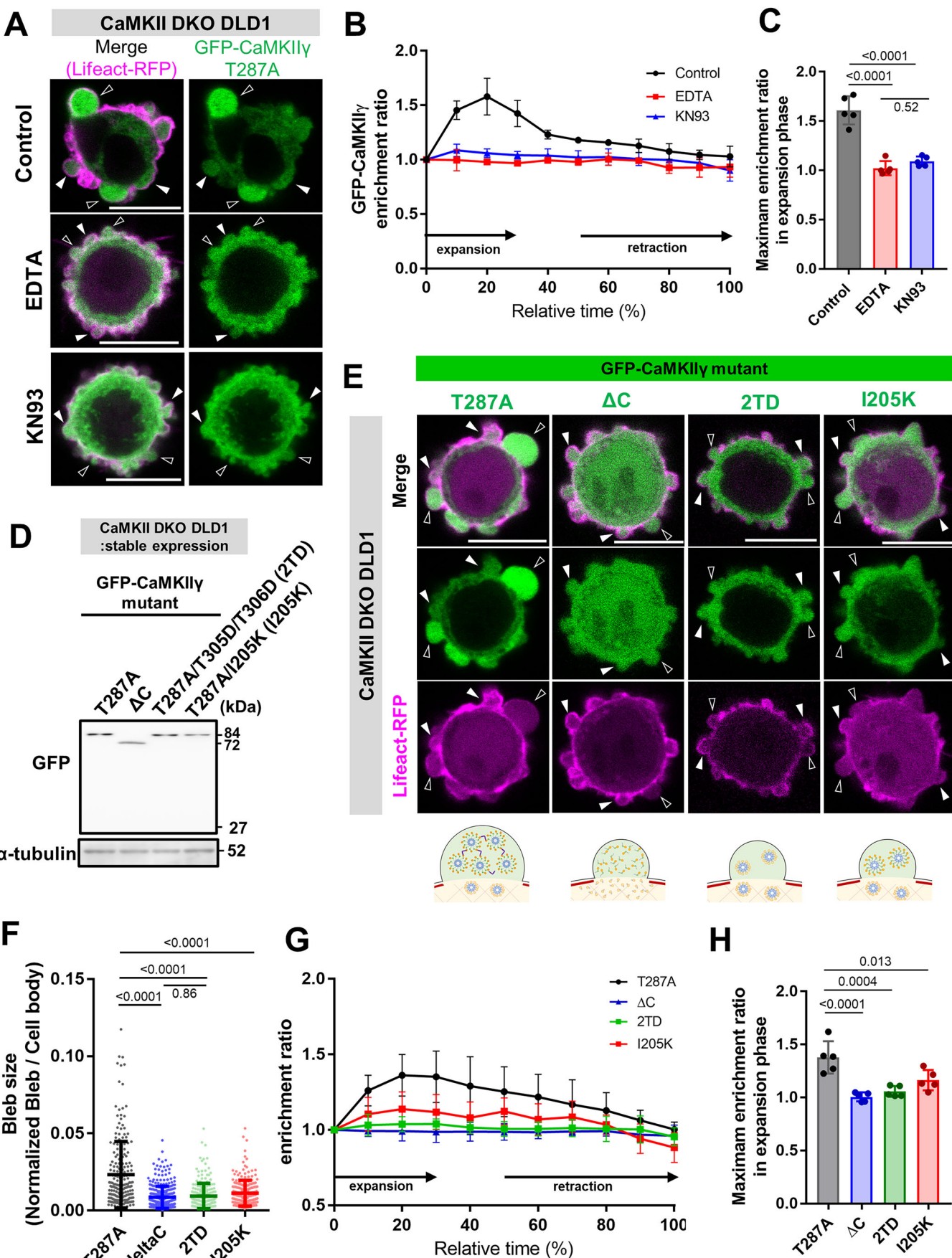

**Figure 5. CaMKII complexes accumulate within blebs during the expansion phase, thereby promoting bleb growth.**

(A) Fluorescence microscopy images of GFP-CaMKIIγ T287A and Lifeact-RFP expressed in CaMKII DKO cells. Treatment with EDTA or KN93 resulted in reduced bleb size and the absence of CaMKII accumulation within blebs. (B) Temporal changes in CaMKII enrichment levels within blebs of cells treated with various reagents. Each dataset represents quantification of five blebs, and the mean ± SD is plotted for each time point. (C) Maximum enrichment ratio of GFP–CaMKIIγ T287A relative to monomeric RFP within expanding blebs of cells treated with the indicated reagents. Quantification was performed for five blebs per group, and statistical significance was evaluated using one-way ANOVA followed by Tukey's post hoc test; $p$ values are $p < 0.0001$ (for Ctrl vs. EDTA), $p < 0.0001$ (for Ctrl vs. KN93), and $p = 0.5229$ (for EDTA vs. KN93). (D) Western blot analysis of CaMKII DKO DLD1 cells stably expressing GFP-CaMKIIγ mutants. α-Tubulin was used as a loading control. (E) GFP-tagged CaMKII mutants and Lifeact-RFP expressed in CaMKII DKO cells. Black arrowheads indicate expanding blebs, while white arrowheads indicate retracting blebs (Scale bar: 10 μm). The schematic below each fluorescence microscopy panel illustrates the bleb size in cells expressing each mutant, the structural features of the mutants, and their subcellular localization. (F) Distribution of bleb sizes in CaMKII DKO cells expressing various CaMKII mutants. GFP expression CaMKII DKO cells were used as a control and labeled as GFP. The areas of all blebs in 20 cells per group were quantified. Results are shown as mean of four biological replicates ± SD and the $p$ values of one-way ANOVA followed by Tukey's post hoc test are indicated; $p$ values are $p < 0.0001$ (for T287A vs. deltaC), $p < 0.0001$ (for T287A vs. 2TD), $p < 0.0001$ (for T287A vs. I205K). (G) Temporal changes in the enrichment rates of various CaMKII mutants in CaMKII DKO cells. Quantification was performed on five blebs per each mutant. Each dataset represents quantification of five blebs, and the mean ± SD is plotted for each time point. (H) Maximum enrichment ratio of GFP–CaMKII mutants relative to monomeric RFP within blebs of cells expressing each mutant. Quantification was performed for five blebs per group, and statistical significance was assessed using one-way ANOVA followed by Tukey's post hoc test; $p$ values are $p < 0.0001$ (for T287A vs. deltaC), $p = 0.0004$ (for T287A vs. 2TD), $p = 0.013$ (for T287A vs. I205K). Source data are available online for this figure.

diffusion in the cytoplasm typically prevents the formation of stable, localized concentration gradients.

A recent study showed that imbalances in protein composition between the interior and exterior of autophagosomes can generate osmotic pressure differences, driving morphological changes such as tubular deformation. The diameter of the autophagosome's aperture is stabilized by Atg24 (Kotani et al, 2023). In yeast lacking Atg24, the aperture of the autophagosome becomes smaller, preventing the internalization of large protein complexes, such as ribosomes and proteasomes, which have a size exceeding 25 nm in diameter (Kotani et al, 2023). Notably, autophagosomes with reduced apertures incapable of incorporating large macromolecules tend to adopt tubular rather than spherical morphologies. Mathematical modeling suggests that even a concentration difference of just a few micromolar between the autophagosome interior and exterior is sufficient to induce such shape changes (Kotani et al, 2023). Given that ribosomes are present in the cytoplasm at concentrations of approximately 10 μM, these differences are biologically plausible. An important consideration is that autophagosomes lack cytoskeletal support that define their shape. The flexible lipid bilayer of autophagosomes, unencumbered by cytoskeletal constraints, can deform in response to small concentration gradients and the resulting osmotic pressure differences. Crucially, the expanding bleb likewise lacks the actin cytoskeleton and thereby presents a highly flexible and deformable membrane architecture. Therefore, we examined the possibility that concentration gradients of CaMKII-based complexes generate sufficient osmotic pressure to expand the bleb membrane.

In the following analysis, we used a mathematical model to determine whether the differences in bleb size during expansion between WT and CaMKII DKO cells could be attributed to osmotic pressure driven by the protein concentration gradient of CaMKII (Fig. 6J,K). Based on the previous work (Tinevez et al, 2009), the bleb was considered to reach its maximum volume at mechanical equilibrium, when the cytoplasmic pressure inside the expanding bleb, $P_{bleb}$, was equal to the cell body pressure, $P_{body}$. The cytoplasmic pressure within the cell body was given by the Laplace pressure imposed by cortical tension minus the elastic resistance of the cell body. On the other hand, the cytoplasmic pressure in the bleb was given by the sum of the Laplace pressure imposed by membrane tension, $P_{tens} = 2\gamma/r_{bleb}$, and the osmotic pressure

difference due to CaMKII complex accumulation in the bleb, $\Delta P_{osmo}$:

$$P_{bleb} = P_{tens} + P_{osmo} = \frac{2\gamma}{r_{bleb}} + \Delta P_{osmo},$$

where $\gamma$ and $r_{bleb}$ were the membrane tension and the maximum bleb radius. Mechanical equilibrium is achieved when the two pressures are equal, $P_{bleb} = P_{body}$.

Since no significant changes were observed in the cell body volume, the accumulation of Myosin II in the actin cortex, or the level of phosphorylated myosin activation between WT and CaMKII DKO DLD1 cells (Appendix Fig. S3 and Appendix Table S1), we concluded that there is no difference in cell body pressure, $P_{body}$, i.e., surface tension and elastic resistance, between the two cell types.

We then estimated the osmotic pressure difference needed to explain the difference in bleb size between WT cells ($r_{bleb}^{WT} \approx 2.8$ μm) and CaMKII DKO cells ($r_{bleb}^{DKO} \approx 1.8$ μm). Assuming the membrane tension, $\gamma = 40$ pN/μm (Tinevez et al, 2009) and no change in cell body pressure, $P_{body}$, between WT and CaMKII DKO cells, the osmotic pressure required to account for the observed difference in bleb size is

$$\Delta P_{osmo} = \frac{2\gamma}{r_{bleb}^{DKO}} - \frac{2\gamma}{r_{bleb}^{WT}} \approx 16 \, \text{pN/μm}^2 = 16 \, \text{Pa}.$$

The osmotic pressure, $\Delta P_{osmo}$, was calculated from the concentration difference, $\Delta c$, using the van't Hoff equation, $\Delta P_{osmo} = \Delta c N_A k_B T$ with the Avogadro number $N_A$, the Boltzmann constant $k_B$ and temperature $T$. The concentrations of CaMKIIγ, CaMKIIδ, Mena, VASP and ERK1 in both the expanding bleb and the cell body were calculated using the method shown in Appendix Fig. S4 (Appendix Tables S1-1 and S1-2). From this, we calculate difference in protein concentration that could generate the osmotic pressure required to account for the difference in bleb size between WT and CaMKII DKO cells (16 Pa) to be 6.6 μM, which is in the same order as the estimated difference in protein concentrations between bleb and non-bleb cytoplasm (3.6 ± 1.7 μM, Appendix Tables S1–2). Therefore, we concluded that the osmotic pressure arising from the protein enrichment within the blebs adequately explains the variation in bleb size between WT and CaMKII DKO

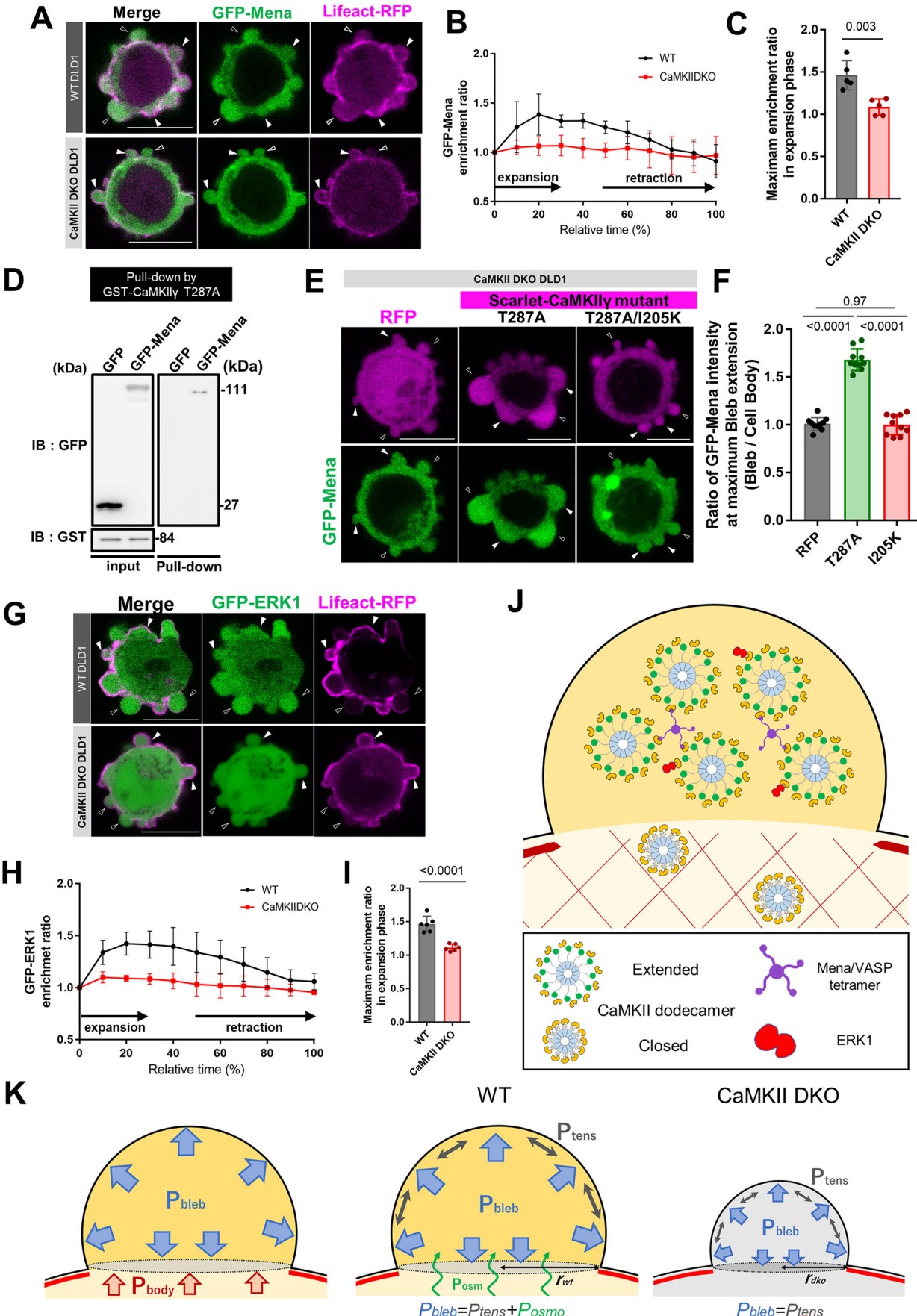

**Figure 6.  Loss of specific protein accumulation within expanding blebs in CaMKII DKO cells.**

(A) GFP-Mena and Lifeact-RFP expressed in WT and CaMKII DKO cells. Black arrowheads indicate expanding blebs, while white arrowheads indicate retracting blebs. In WT cells, GFP-Mena accumulates in expanding blebs, whereas this accumulation is lost in CaMKII DKO cells (Scale bar: 10 μm). (B) Temporal changes in the enrichment rates of GFP-Mena within blebs in WT and CaMKII DKO DLD1 cells. Each dataset represents quantification of five blebs, and the mean ± SD is plotted for each time point. (C) Maximum enrichment ratio of GFP–Mena relative to monomeric RFP within expanding blebs of wild-type and CaMKII DKO cells. Quantification was performed for five blebs per group, and statistical significance was evaluated using Student's t-test; $p$ value is $p = 0.0028$. (D) Pulldown assay using glutathione beads coated with GST-hCaMKIIγ T287A and lysates from HEK293 cells expressing GFP-Mena or GFP. (E) GFP-Mena and mScarlet-tagged CaMKII mutants expressed in CaMKII DKO cells. (F) Fluorescence intensity ratio of GFP–Mena (bleb/cell body) at the point of maximal bleb expansion in CaMKII DKO cells co-expressing mScarlet-tagged CaMKII mutants and GFP–Mena. Quantification was performed for ten blebs per group, and statistical significance was assessed using one-way ANOVA followed by Tukey's post hoc test; $p$ values are $p < 0.0001$ (for RFP vs. T287A), $p < 0.0001$ (for T287A vs. I205K), $p = 0.9728$ (for RFP vs. I205K). (G) GFP-ERK1 and Lifeact-RFP expressed in WT and CaMKII DKO cells. Black arrowheads indicate expanding blebs, while white arrowheads indicate retracting blebs. In WT cells, GFP-ERK1 accumulates in expanding blebs, whereas this accumulation is lost in CaMKII DKO cells (Scale bar: 10 μm). (H) Temporal changes in the enrichment rates of GFP-ERK1 within blebs in WT and CaMKII DKO cells. Each dataset represents quantification of five blebs, and the mean ± SD is plotted for each time point. (I) Maximum enrichment ratio of GFP–ERK1 relative to monomeric RFP within expanding blebs of wild-type and CaMKII DKO cells. Quantification was performed for five blebs per group, and statistical significance was assessed using Student's t-test; $p$ value is $p < 0.0001$. (J) Proposed cytoplasmic environment of expanding blebs, integrating our experimental observations. High local $Ca^{2+}$ concentrations within the bleb favor CaMKII dodecamer activation, leading to the formation of large protein complexes through multivalent interactions with other multimeric proteins such as Mena (shown in blue). Proteins capable of interacting with CaMKII, such as ERK1, may also become embedded in this large, interconnected network, collectively promoting bleb expansion. (K) Schematic of the mathematical model. Source data are available online for this figure.

cells, which we term CODE (C̲aMKII-based O̲smotically-driven D̲eformation).

## The structural changes in CaMKII and its accumulation within blebs are essential for amoeboid cell migration

DLD1 cells form multiple blebs in random directions, whereas cells utilizing blebs for locomotion form a single bleb at the leading edge in the direction of movement. A pioneering study observed the formation of blebs during the migration of primordial germ cells (PGCs) within zebrafish embryos, a process referred to as amoeboid migration. Interestingly, intracellular $Ca^{2+}$ level is elevated in the expanding blebs of PGCs in response to stimuli such as chemokine receptor activation (Blaser et al, 2006). However, the relationship between intracellular $Ca^{2+}$ levels and cell motility-associated blebs is unknown. Therefore, we examined whether the CaMKII-mediated mechanism of bleb expansion is also involved in amoeboid migration.

Walker256 cells, a rat breast cancer cell line, exhibit amoeboid migration by forming blebs at the leading edge in a three-dimensional environment with limited adhesive substrates (Bergert et al, 2012). First, we investigated whether $Ca^{2+}$ levels increase within the cytoplasm of expanding blebs in Walker256 cells undergoing amoeboid migration. Walker256 cells expressing GCaMP6s were embedded in type I collagen gels and observed via live imaging. Fluorescence intensity analysis revealed that $Ca^{2+}$ levels were elevated specifically within blebs at the leading edge in the direction of migration (Fig. 7A,B). GFP-CaMKIIγ accumulated within the expanding blebs at the leading edge (Fig. 7C,D and Movie EV3).

Next, we used Camui-CR to assess the structural state of CaMKII during amoeboid migration. Live imaging showed that in leading-edge blebs, Camui-CR underwent a structural transition to the activation-competent extended conformation, indicating structural activation of CaMKII (Fig. 7E). These results are consistent with findings from DLD1 cell bleb analyses, suggesting that the localized elevation of cytoplasmic $Ca^{2+}$ and the resulting structural changes and accumulation of CaMKII also occur within directional blebs during amoeboid migration.

To evaluate the impact of CaMKII on amoeboid migration, Walker256 cells embedded in type I collagen gels were treated with KN93, an inhibitor of CaMKII structural changes. KN93-treated cells exhibited a marked reduction in the size of blebs formed at the leading edge of polarized cells (Fig. 7F,G). KN93-treated Walker256 cells exhibited a significant reduction in migration speed during amoeboid movement (Fig. 7H,I). Taken together, these findings suggest that the mechanism of CaMKII-mediated bleb expansion plays a critical role in facilitating amoeboid migration.

## Discussion

In this study, we elucidated a novel role of CaMKII in the mechanism of bleb formation during cell migration. Specifically, we demonstrated that CaMKII concentrates in the cytoplasm of expanding blebs, generating osmotic pressure that facilitates fluid influx, thereby contributing to the expansion of blebs which lack actin cortex. We propose that CaMKII functions as a physical scaffold regulating dynamics of cytoplasm.

CaMKII is a critical regulator of spine formation in neuronal cells (Lisman et al, 2002), with its function attributed either to kinase activity or structural changes. In this study, we revealed that CaMKII dodecamers form complexes exceeding the pore size of the actin meshwork, creating localized cytoplasmic concentration differences that generate osmotic pressure. This osmotic activity facilitates fluid influx, which causes the expansion of bleb membranes that are susceptible to deformation, due to the absence of actin structures. A recent study on autophagosome formation suggested that specific proteins could exhibit osmotic activity in cells. Atg24 mutants have been shown to induce deformation of autophagosomes via ribosomal concentration gradients (Kotani et al, 2023). Our findings are significant as they extend this concept to physiological scenarios, providing the first demonstration of protein-based osmotic pressure-driven membrane deformation in cellular migration.

The accumulation of CaMKII within leading-edge blebs during amoeboid migration suggests that it may contribute to amoeboid motility in diverse cell types, such as neutrophils, primordial germ cells (PGCs), and cancer cells. High expression of CaMKII have

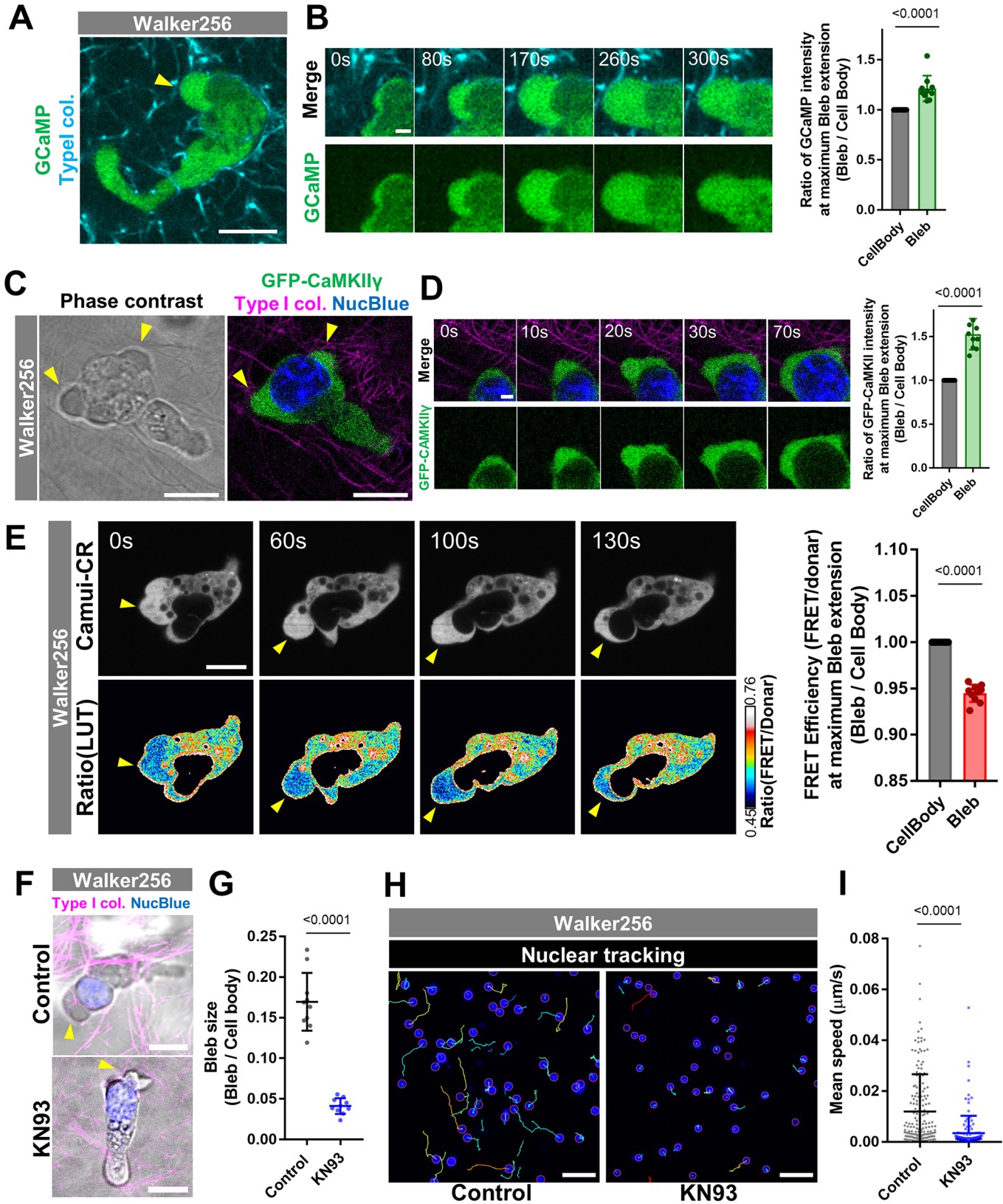

**Figure 7. Ca²⁺-dependent structural change of CaMKII is essential for amoeboid movement.**

(A) Amoeboid movement of Walker256 cells expressing GCaMP6s in type I collagen gel. Overview of a moving cell. Increased fluorescence intensity of GCaMP6s is observed in the leading-edge bleb (yellow arrowhead) (Scale bar: 10 μm). (B) Time-lapse images of a bleb. Increased fluorescence intensity of GCaMP6s is observed in expanding blebs (Scale bar: 2 μm). Fluorescence intensity ratio between leading-edge bleb and cell body regions during bleb expansion (right panel). Quantification was performed for ten blebs and statistical significance was assessed using Student's t-test; $p$ value is $p < 0.0001$. (C) Amoeboid movement of Walker256 cells expressing GFP-CaMKIIγ in type I collagen gel. Overview of a moving cell. Accumulation of GFP-CaMKIIγ is observed in the leading-edge bleb (yellow arrowhead) (Scale bar: 10 μm). (D) Time-lapse images of a bleb. Increased fluorescence intensity of GFP-CaMKIIγ is observed in expanding blebs (Scale bar: 2 μm). Fluorescence intensity ratio between leading-edge bleb and cell body regions during bleb expansion (right panel). Quantification was performed for ten blebs and statistical significance was assessed using Student's t-test; $p$ value is $p < 0.0001$. (E) Amoeboid movement of Walker256 cells expressing the FRET probe Camui-CR in type I collagen gel. Top: Distribution of Camui-CR. Bottom: Pseudocolored FRET ratio images. In persistent blebs at the leading edge (yellow arrowhead), a decrease in the FRET ratio is observed (Scale bar: 10 μm). Relative FRET efficiency between leading-edge bleb and cell body regions during bleb expansion (right panel). Quantification was performed for ten blebs and statistical significance was assessed using Student's t-test; $p$ value is $p < 0.0001$. (F) Effect of CaMKII inhibition on bleb formation in migrating Walker256 cells. Addition of KN93 to Walker256 cells undergoing amoeboid migration within type I collagen resulted in smaller blebs at the leading edge (yellow arrowheads) compared with control cells (Scale bar: 10 μm). (G) Quantification of bleb size at the leading edge of Walker256 cells undergoing amoeboid migration within type I collagen. We quantified bleb size by measuring the cross-sectional areas of the bleb and the cell body at the moment when each bleb reached its maximum expansion at the leading edge of control Walker256 cells and Walker256 cells treated with KN-93. The ratio of the bleb area to the cell-body area was then calculated for comparison. Quantification was performed for ten blebs. Results are shown as mean of four biological replicates ± SD and the $p$ value of Student's t-test is indicated; $p$ value is $p < 0.0001$. (H) Tracking results of Walker256 cell nuclei during movement in type I collagen gel. Cells treated with KN93 exhibit significantly reduced motility. (I) Quantification of mean nuclear displacement velocity among cells under different conditions. Results are shown as mean of three biological replicates ± SD and the $p$ value of Student's t-test is indicated; $p$ value is $p < 0.0001$. Source data are available online for this figure

been reported to promote cancer cell migration in various cancers, including breast cancer (Britschgi et al, 2013) and prostate cancer (Wang et al, 2014), supporting this relationship. Of particular interest, melanoma cells are known to utilize blebs for invasion (Sahai and Marshall, 2003; Ullo and Logue, 2021). Investigating whether CaMKII structural changes, targeted by inhibitors such as KN93, can suppress cancer invasion represents a promising avenue for future research. Additionally, blebs are implicated in providing anoikis resistance to cancer cells (Weems et al, 2023). We found that ERK1, a key regulator of cell proliferation and apoptosis suppression, accumulates in the cytoplasm of expanding blebs in a CaMKII-dependent manner. Interestingly, Raf-1, an upstream activating kinase of ERK1, is known to directly interact with CaMKII (Illario et al, 2003; Salzano et al, 2012). This interaction was shown to modulate Raf-1 activation and downstream ERK signaling, suggesting the possibility that CaMKII may function as a scaffold that organizes multiple components of the MAPK pathway. However, neither Raf-1 nor MEK accumulated in expanding blebs (Fig. EV2), indicating that ERK1 accumulation at the bleb cortex occurs independently of the canonical Raf–MEK cascade. These observations raise the intriguing possibility that CaMKII and ERK1 form a local, non-canonical signaling module. From the perspective of the role of blebs in conferring anoikis resistance to cancer cells, functional analysis of the cohort of proteins that accumulate in blebs via CaMKII scaffolding could provide valuable insights into potential therapeutic strategies targeting anoikis resistance.

Our findings suggest that CaMKII functions not only as a kinase but also as a scaffold regulating local osmotic pressure and membrane deformation. This novel mode of action we call CODE (CaMKII-based Osmotically-driven Deformation) should provide insight into other calcium-triggered membrane deformation phenomena, such as dendric spine expansion (Matsuzaki et al, 2004) and micro vesicle release (Smalheiser, 2007).

Recent studies have shown that CaMKII can undergo liquid–liquid phase separation (LLPS) with the NMDA receptor subunit GluN2B, forming biomolecular condensates proposed to organize synaptic signaling complexes (Brown and Bayer, 2024; Cai et al, 2021; Hosokawa et al, 2021). While such neuronal condensates have been linked to synaptic plasticity (Cai et al, 2021; Hosokawa et al, 2021),

their physiological relevance remains under debate (Rumian et al, 2024). In contrast, the CaMKII assemblies identified in our study exhibit a clearly defined mechanical function: they generate osmotic pressure that drives membrane deformation during bleb expansion. Notably, the sustained elevation of cytoplasmic Ca²⁺ within expanding blebs maintains CaMKII in an activation-competent conformation, eliminating the need for autophosphorylation at T287, which is required for persistence of neuronal condensates after transient Ca²⁺ spikes (Cai et al, 2021; Hosokawa et al, 2021). These observations suggest that the principle of CaMKII-driven molecular condensation can manifest in distinct physiological contexts—synaptic organization in neurons and mechanical deformation in motile cells—depending on the dynamics of Ca²⁺ signaling. Thus, we propose that CaMKII-based supramolecular complexes represent a functional form of biomolecular condensation that directly modulates cellular mechanics.

Future research should focus on further elucidating the molecular mechanisms underlying CaMKII structural changes and evaluating its potential as a therapeutic target for inhibiting cancer invasion and overcoming anoikis resistance. Advancing these studies will deepen our understanding of CaMKII-mediated cellular phenomena and pave the way for the development of novel therapeutic strategies.

## Methods

**Reagents and tools table**

| Reagent/Resource | Reference or Source | Identifier or Catalog Number |
|---|---|---|
| **Experimental models** | | |
| DLD1 | ATCC | CCL-221 |
| HEK293T | ATCC | CRL-3216 |
| Walker256 | Gift from Dr. Jolana Sroka | |
| **Recombinant DNA** | | |
| pGP-CMV-GCaMP6s | Addgene | Cat # 40753 |
| pcDNA3-Camui-CR | Addgene | Cat # 40256 |

| Reagent/Resource | Reference or Source | Identifier or Catalog Number |
|---|---|---|
| pSems LifeAct-StayGold | Addgene | Cat # 222947 |
| pLenti-CRISPR v2 | Addgene | Cat # 52961 |
| pLenti GFP-CaMKIIγ | In this study | N/A |
| pLenti GFP-CaMKIIγ T287A | In this study | N/A |
| pLenti GFP-CaMKIIγ T287A K43M | In this study | N/A |
| pLenti GFP-CaMKIIγ delta C | In this study | N/A |
| pLenti GFP-CaMKIIγ T287A T305D T306D | In this study | N/A |
| pLenti GFP-CaMKIIγ T287A I205K | In this study | N/A |
| pLenti CRISPR v2 human CaMKIIγ | In this study | N/A |
| pLenti CRISPR v2 human CaMKIIδ | In this study | N/A |
| **Antibodies** | | |
| Rabbit anti-CaMKII gamma | Proteintech | #12666-2-AP |
| Mouse anti-CaMKII delta | Abcam | #EPR13095 |
| Rabbit anti-Phospho-p70 S6 Kinase (Thr389) | Cell Signaling Technology | #9205 |
| Rabbit anti-Ribosomal Protein S3 Antibody | Cell Signaling Technology | #2579 |
| Mouse anti-α-tubulin | In house | 12G10 |
| Rat anti-GFP | In house | JFP-5 |
| Goat Anti-Rabbit IgG-HRP | Southern Biotech | #4030-05 |
| Goat anti-Mouse IgG-HRP | Bethyl Laboratories | #A90-516P |
| **Oligonucleotides and other sequence-based reagents** | | |
| CRISPR target sequence for CaMKIIγ (human) 5´-GTGCTGATCCTCATCCCAGA-3´ | This study | |
| CRISPR target sequence for CaMKIIδ (human) 5´-GAGGCTGTGATGCGTTTGGC-3´ | This study | |
| **Chemicals, Enzymes and other reagents** | | |
| AccuDia D-MEM 2 | Shimadzu Diagnostics Corporation | 05919 |
| AccuDia RPMI1640 | Shimadzu Diagnostics Corporation | 05911 |
| Leibovitz's L-15 Medium | Thermo Fisher Scientific | 11415064 |
| PEI MAX | Polysciences | 24765-1 |
| AnCoA4 | Calbiochem | 5.32999.0001 |
| SKF96365 | Tocris Bioscience | 1147 |
| Rapamycin | LC Laboratories | 53123-88-9 |
| Alexa Fluor 488 Phalloidin | Thermo Fisher Scientific | A12379 |
| Can Get Signal immunostain Immunoreaction Enhancer Solution Solution B | TOYOBO | NKB-601 |
| paraformaldehyde | Nacalai Tesque | 26126-54 |
| glutaraldehyde | Nisshin EM | 3041 |

| Reagent/Resource | Reference or Source | Identifier or Catalog Number |
|---|---|---|
| Native Collagen Acidic Solution I-AC | KOKEN | IAC-50 |
| ATTO-647 type I collagen | In this study | N/A |
| **Software** | | |
| Excel for Microsoft 365 MSO 2306 | Microsoft | |
| Prism v8.4.1 | Graphpad | |
| ImageJ/Fiji | | https://imagej.net/software/fiji/ |
| **Other** | | |
| LSM900 with Airyscan2 | Carl Zeiss | |
| Dragonfly200 | OXFORD Instruments | |

## Cells and reagents

DLD1 cells and HEK293 cells were purchased from ATCC. These cell lines were authenticated by short tandem repeat (STR) profiling using the GenePrint10 System (Promega). Walker256 cells was a kind gift from Dr. Jolanta Sroka (Jagiellonian University, Krakow, Poland). All cell lines were routinely tested and confirmed to be negative for mycoplasma contamination. DLD1 cells and HEK293 cells were grown in DMEM supplemented with 10% (vol/vol) fetal calf serum (FCS) (Sigma). Walker256 cells were cultured in RPMI1640 supplemented with 10% (vol/vol) fetal calf serum (FCS) (Sigma). For calcium chelation, DLD1 cells were treated with 5 mM EDTA in calcium-free DMEM for 10 min prior to imaging. The culture medium contained approximately 1.8 mM $Ca^{2+}$ and 0.8 mM $Mg^{2+}$ under control conditions. DLD1 cells were treated with 50 μM AnCoA4 (Calbiochem) and 10 μM SKF96365 (Tocris Bioscience) to inhibit SOCE activity. To reduce the endogenous ribosome level, DLD1 cells were treated with 1 μM rapamycin (LC Laboratories) for 3 h. The following primary antibodies were used for immunoblotting and immunofluorescence: Rabbit anti-CaMKII gamma Ab (#12666-2-AP, Proteintech, 1:1000), mouse anti-CaMKII delta Ab (#EPR13095, Abcam, 1:1000), Rabbit anti-Phospho-p70 S6 Kinase (Thr389) Ab (#9205, Cell Signaling Technology, 1:1000), Rabbit anti-Ribosomal Protein S3 Antibody (#2579, Cell Signaling Technology, 1:1000), mouse anti-α-tubulin (12G10) mAb (1:1000) and Rat anti-GFP (JFP5) Ab (1:50). Secondary antibodies were as follows: Goat Anti-Rabbit IgG-HRP (#4030-05, Southern Biotech, 1:1000); Goat anti-Mouse IgG-heavy and light chain cross-adsorbed Antibody HRP Conjugate (#A90-516P, Bethyl Laboratories, 1:1000). Alexa Fluor 488 Phalloidin (#A12379, 1:100) was purchased from Thermo Fisher Scientific. Following expression vectors were purchased from Addgene; pGP-CMV-GCaMP6s (Addgene No.40753), pcDNA3-Camui-CR (Addgene No. 40256) and pSems LifeAct-StayGold (Addgene No. 222947). cDNAs encoding full-length human CaMKIIγ, human CaMKIIδ, human Mena, human VASP, human ERK1, human MRLC1 and mouse PLCδ-PH were amplified by RT-PCR, fused to the sequence encoding EGFP or mScarlet, and ligated into the pCAGGS-neo vector. Oligonucleotides were phosphorylated, annealed, and cloned into the BsmBI site of pLenti-CRISPR v2

vector (Addgene No.52961) according to Zhang laboratory protocols (F. Zhang, Massachusetts Institute of Technology, Cambridge, MA). The target sequences were as follows:

CaMKIIγ (human), 5′-GTGCTGATCCTCATCCCAGA-3′;
CaMKIIδ (human), 5′-GAGGCTGTGATGCGTTTGGC-3′;

## Gene expression

Transient gene transfection into DLD1 and HEK293 cells was performed using the polyethyleneimine (PEI) method. Briefly, plasmid DNA (final concentration: 1 μg/mL) and PEI MAX (3 μg/mL) were added to DMEM (calcium-free; Gibco) at 5% of the total culture medium volume and mixed thoroughly. The mixture was then incubated at room temperature for 30 min. In a separate dish containing cells, half the total volume of medium was initially used for culture; the DNA–PEI mixture was subsequently added, and the cells were incubated at 37 °C. Four hours later, 10% FBS/DMEM was added to bring the medium to its final volume, and incubation continued at 37 °C. All subsequent experiments were conducted 24–48 h post-transfection.

For generating CaMKII knockout cell lines, we utilized lentiviral vector-mediated gene transfer. Lentiviral vectors were produced by transfecting HEK293 cells with plasmids for lentivirus production, along with psPAX2 (Addgene No. 12260) and pMD2.G (Addgene No. 12259), using the PEI method. After 48 h, the culture medium was collected and centrifuged at 15,000 rpm for 5 min at room temperature; the resulting supernatant was used as the viral solution. CaMKII knockout cells were generated via the CRISPR-Cas9 system. For CaMKIIγ knockout, infected cells were selected and cloned using hygromycin (final concentration: 100 μg/mL; Invitrogen). For CaMKIIδ knockout, G-418 (final concentration: 500 μg/mL; Nacalai Tesque) was used to establish knockout cell lines.

To generate CaMKII DKO cell lines stably expressing GFP-tagged CaMKII mutants, viral supernatants were produced using HEK293 cells as described above and used to infect the previously established CaMKII DKO DLD1 cells. Infected cells were sorted using a Cell Sorter SH800S (SONY), and only cells with GFP fluorescence intensities between $1.0 \times 10^5$ and $2.0 \times 10^5$ were collected and cultured. Subsequently, a second round of sorting was performed to collect cells with GFP fluorescence intensities between $1.00 \times 10^5$ and $1.05 \times 10^5$ for further culture. The expression levels and molecular weights of the GFP-tagged proteins were confirmed by Western blotting using an anti-GFP antibody.

Transient transfection of Walker256 cells was carried out via electroporation. A total of $1 \times 10^6$ cells were suspended in 100 μL of plasmid mixture containing 10 μg of plasmid DNA in Opti-MEM (Life Technologies). Electroporation was performed using a NEPA21 electroporator (Nepa Gene) and the NEPA Cuvette electrode (Nepa Gene) under the following conditions: two poring pulses at 175 V for 2.5 ms each, five transfer pulses at 20 V for 50 ms each, with a 50 ms interval between pulses. Immediately after electroporation, cells were transferred into 10% FBS/RPMI1640 and incubated for 24–48 h at 37 °C in a 5% $CO_2$ environment.

## Live imaging

Fluorescence imaging was performed using a 63×oil-immersion objective on an inverted microscope (LSM900; Carl Zeiss MicroImaging) interfaced to a laser-scanning confocal microscope equipped with a heating stage heated to 37 °C. Nuclei was stained using NucBlue Live Ready Probe Reagent (Invitrogen). Images were captured on a device camera and acquired using Carl Zeiss Zen 3.4 software. Images were acquired at 488 nm for GFP-tagged proteins or at 555 nm for RFP, mCherry or mScarlet-tagged proteins. Each imaging video frame is a 16-bit grayscale image, and the frame interval is indicated in the supplementary movie legends.

DLD1 cells spontaneously formed blebs when seeded at a density of $1 \times 10^6$ cells in an uncoated 35-mm glass-bottom dish after about 6 h. For live imaging of Walker256 cells, a type I collagen gel (2.5 mg/mL) containing Walker256 cells at $5 \times 10^6$ cells/mL was polymerized in a 35-mm glass-bottom dish. RPMI medium was then added, and the cells were incubated at 37 °C under 5% $CO_2$ for 30 min. Subsequently, the medium was replaced with 10% FBS/RPMI 1640 (containing HEPES; Wako) prewarmed to 37 °C, and the cells were maintained at 37 °C using a stage heater throughout observation. For nuclear labeling, NucBlue Live Ready Probe Reagent (Invitrogen) was added to the medium. Type I collagen gels were prepared according to the manufacturer's instructions for the collagen acidic solution I-PC (KOKEN). In certain experiments, 10% of the total collagen content was replaced with Atto 647-labeled type I collagen synthesized in our laboratory, and the resulting gels were used for imaging.

## FRET imaging

FRET imaging was performed using a ×100 oil-immersion objective on an inverted microscope (IX83; Olympus corporation) interfaced to a spinning-disk confocal microscopy (Dragonfly200; OXFORD Instruments) equipped with a heating stage heated to 37 °C. Images of cells expressing Camui-CR were excited at 488 nm, and the resulting fluorescence was separated into two parts using a low-pass filter at 560 nm, which were captured simultaneously using two device cameras and acquired using Fusion software (Dragonfly200; OXFORD Instruments).

## Super-resolution imaging

To capture snapshot images of the fine actin structures present in blebbing cells, we used an LSM900 super-resolution microscope (Carl Zeiss) equipped with Airyscan2. A 63× oil-immersion objective lens (PlanApo 63x/1.40 NA Oil) was employed. The imaging mode was set to AiryScan SR, and data were acquired without averaging. Images were then processed with the Airyscan software according to the manufacturer's instructions, following the software's automatically determined settings.

## Immunofluorescence

To compare bleb sizes, cells expressing mScarlet-mPLCδPH cultured in glass-bottom dishes were fixed with 4% paraformaldehyde (PFA) for 15 min at 37 °C and washed once with PBS for 10 min. Fluorescent immunostaining was performed to observe myosin phosphorylation levels in each cell. Cells cultured in glass-bottom dishes were fixed with 4% PFA for 15 min at 37 °C and permeabilized with 0.1% Triton X-100 prepared in PBS. Fixed cells were blocked with 1% BSA prepared in PBS for 1 h at RT. Cells were incubated with primary antibodies for 1 h at RT and

secondary antibodies for 1 h at RT. Antibodies were prepared in the blocking solution.

For immunofluorescence analysis of endogenous CaMKII localization in fixed cells, wild-type DLD1 cells cultured on glass-bottom dishes were fixed with a mixture of 4% paraformaldehyde and 1% glutaraldehyde at 37 °C for 15 min, followed by washing with PBS. Cross-linking activity of the fixative was quenched with 0.1 M glycine prepared in PBS at room temperature for 10 min and permeabilized with 0.1% Triton X-100 prepared in PBS. Fixed cells were blocked with 1% BSA prepared in PBS for 1 h at room temperature. Primary and secondary antibodies were diluted in Can Get Signal immunostain solution B (TOYOBO) and incubated with the cells for 1 h at room temperature.

Fixed cells were observed at RT. All observation was performed with a confocal microscope (Carl Zeiss LSM900) equipped with Plan-APO (63×/1.40 NA, oil immersion) objective. Images were acquired using Carl Zeiss Zen 3.4 software. Images were analyzed using ImageJ/Fiji.

## Kinase assay of CaMKII mutants

The kinase activity of GFP-tagged CaMKIIγ autophosphorylation-deficient (T287A) and kinase-inactive (T287A/K43M) mutants was quantified using a CaM kinase II Assay Kit (Cylex). CaMKII DKO cells stably expressing GFP only (negative control), CaMKIIγ WT, T287A, or T287A/K43M were seeded at $5 \times 10^5$ cells per well in 6-well plates and cultured at 37 °C in a 5% $CO_2$ incubator for 24 h. After incubation, the cells were washed with ice-cold PBS and lysed in 1 mL of lysis buffer containing protease inhibitors, followed by incubation on ice for 5 min. Lysates were collected into 1.5-mL tubes and centrifuged at 15,000 rpm for 10 min at 4 °C. The supernatants were diluted 10-fold with lysis buffer and used as CaMKII-containing samples. The subsequent assay steps were performed according to the manufacturer's instructions. Absorbance was measured at 450 nm and 595 nm using an ARVO X3 microplate reader (PerkinElmer). Absorbance values obtained from GFP-expressing CaMKII DKO cell-derived samples were used as the baseline and subtracted from each measurement to calculate the net absorbance corresponding to CaMKII-dependent kinase activity. The kinase activity of wild-type CaMKII was set to 1, and all values were normalized accordingly to yield the relative kinase activity.

## Pull-down assay

HEK293 cells expressing target protein were washed with ice-cold PBS and lysed with cell lysis buffer (20 mM Tris-HCl [pH 7.5], 150 mM NaCl, and 1% Triton X-100) supplemented with 10 µg/ml leupeptin (334-40414; Wako Pure Chemical Industries), 2 µg/ml aprotinin (10236624001; Roche), and 50 µM amidinobenzylsulfonyl fluoride (015-26333; Wako Pure Chemical Industries). Clarified lysates were incubated with GST-hCaMKIIγ T287A bound to glutathione Sepharose beads at 4 °C for 2 h. Beads were washed with the lysis buffer, and bound proteins were dissolved in SDS sample buffer.

## Immunoblotting

Whole cells were lysed with SDS sample buffer and samples were resolved by SDS-PAGE and transferred to nitrocellulose

membranes. After blocking with 5% BSA prepared in TBS-T, membranes were incubated with primary antibody for 1 h at RT or overnight at 4 °C. Membranes were then washed and incubated with HRP-conjugated secondary antibody for 1 h at RT. Chemiluminescence signal was detected using a LAS4000mini imaging system (Fujifilm) and images were analyzed using ImageJ/Fiji.

## Estimation of the absolute concentration of a particular protein in blebs versus the cell body

We first purified each GST-tagged protein intended for quantification and determined its concentration. Using whole-cell lysates from a known number of cells, we then calculated the amount of protein per cell. WT cells and CaMKII DKO cells expressing GFP-tagged protein of interest were subsequently fixed with 4% PFA for 15 min at 37 °C and imaged in three dimensions by confocal microscopy at 0.2-µm slice intervals. Regions of interest (ROIs) corresponding to the cytoplasmic cell body (excluding the nucleus) and each bleb were manually selected, and their volumes and average fluorescence intensities were quantified using ImageJ/Fiji. From these measurements, the relative abundance of the target protein was determined for each region. Finally, by integrating the total cellular protein content (as derived from the purified GST-tagged protein standards), we calculated both the average protein concentration in the cell and the differential protein concentrations between the cell body and individual blebs.

## Quantification of protein enrichment in the cytoplasm of expanding blebs

To quantify the degree of protein concentration within the cytoplasm of blebs during their expansion phase, cells co-expressing GFP-tagged protein of interest and monomeric RFP (an internal reference fluorescent marker) were subjected to live imaging. Using ImageJ/Fiji, regions of interest (ROIs) were manually defined for the bleb and the cell body in the resulting time-lapse images. Specifically, the frame at which bleb formation was first confirmed was set as t = 1, and the analysis was carried out from one frame prior to bleb formation (t = 0) until the bleb retracted and disappeared. Fluorescence intensities in each region were measured separately for GFP and RFP at every time point. The ratio of fluorescence intensity in the bleb region to that in the cell body region was then calculated for each color channel. Subsequently, the ratio of GFP to RFP fluorescence intensity (i.e., the GFP/RFP ratio) was used to define the "concentration factor," and its temporal changes were quantified. Based on the quantitative data obtained, relative time was calculated from the total number of frames from the onset of bleb formation to the completion of retraction. The data were linearly interpolated using the TREND function in Excel and used for subsequent analyses. To compare the degree of enrichment of each GFP-tagged protein within expanding blebs, the maximum enrichment ratio of GFP relative to monomeric RFP observed during the period from bleb initiation to maximal expansion was analyzed.

## Quantification of bleb size

Bleb sizes were quantified using the outline of fluorescent proteins, either mScarlet-PLCδPH in WT and CaMKII DKO cells or the GFP-

tagged CaMKII mutants in the CaMKII DKO rescue cells. $1 \times 10^6$ cells were seeded in a 35-mm glass-bottom dish and incubated for 6 h at 37 °C under 5% $CO_2$. The medium was then completely removed, and the cells were fixed by adding 8% PFA prewarmed to 37 °C, followed by incubation for 20 min at 37 °C. Random fields of view were selected, and images of a 202.83 µm × 202.83 µm area were acquired at 4096 × 4096 pixels. Within each field, 10–25 cells were randomly chosen for bleb measurements. Using ImageJ/Fiji, ROIs were defined for all visible blebs and for the cell body, and the area of each ROI was measured. The bleb area was then normalized to the total cell body area. This procedure was repeated for an equal number of cells in each sample.

### Evaluation of cell motility of Walker256 cells

Wild-type Walker256 cells labeled with NucBlue Live Ready Probe Reagent were embedded in 2.5% type I collagen gels under the same conditions used for live imaging. One hour before observation, the medium was replaced with 10% FBS/RPMI 1640 (with HEPES) containing either DMSO or KN93 (final concentration: 20 µM), and cells were maintained at 37 °C. Imaging was carried out using a Plan-APO 40×/0.95 NA oil-immersion lens, capturing a 319.5 µm × 319.5 µm area at 1024 × 1024 pixels. Two averaging scans were performed for each acquisition, and a total of seven z-slices (3 µm apart, covering ~21 µm in total) were collected at 1-minute intervals over a 30-minute period. The resulting z-stack time-lapse images of nuclei were converted into two-dimensional time-lapse images via the Max Intensity Z-projection function in ImageJ/Fiji. Cell nuclei were then tracked and analyzed using the TrackMate plugin. Nuclei were detected as ~5-pixel diameter bright spots using the DoG (Difference of Gaussian) detector, and the Simple LAP tracker algorithm was employed for tracking. Only tracks longer than five frames were retained for subsequent analysis of velocity and distance traveled.

### Statistical analysis

Microsoft Excel for Microsoft 365 MSO 2306 and GraphPad Prism v8.4.1 were used for analyses and displays of quantitative data. Data are expressed as mean with each point or error bars representing standard deviations (SD). Unpaired two-tailed Student's *t*-test, ANOVA with Dunnett's multiple comparison test and Welch's *t*-test are performed using Prism. Unless otherwise noted, *n* denotes biological replicates.

## Data availability

This study includes no data deposited in external repositories.

The source data of this paper are collected in the following database record: biostudies:S-SCDT-10_1038-S44318-026-00703-5.

## Peer review information

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

## Acknowledgements

We thank all members of the Ikenouchi laboratory for helpful discussions. We appreciate the technical assistance from The Research Support Center, Research Center for Human Disease Modeling, Kyushu University Graduate School of Medical Sciences, which is partially supported by the Mitsuaki Shiraishi Fund for Basic Medical Research. This work was supported by AMED-FORCE (21444781) (JI), JSPS KAKENHI [JP25H01325 (JI), JP25H00994 (JI), JP23K05715 (YS) and JP24KJ1767 (YF)], JST-FOREST (JPMJFR204L) (JI), and the Bioscience Research Grant from Takeda Science Foundation (JI).

## Author contributions

**Yuki Fujii:** Resources; Data curation; Formal analysis; Funding acquisition; Validation; Investigation; Visualization; Methodology; Writing—original draft. **Yuji Sakai:** Formal analysis; Funding acquisition; Investigation; Writing—original draft. **Kenji Matsuzawa:** Data curation; Formal analysis; Writing—review and editing. **Junichi Ikenouchi:** Conceptualization; Writing—original draft; Project administration; Writing—review and editing.

Source data underlying figure panels in this paper may have individual authorship assigned. Where available, figure panel/source data authorship is listed in the following database record: biostudies:S-SCDT-10_1038-S44318-026-00703-5.

## Disclosure and competing interests statement

The authors declare no competing interests.

# Expanded View Figures

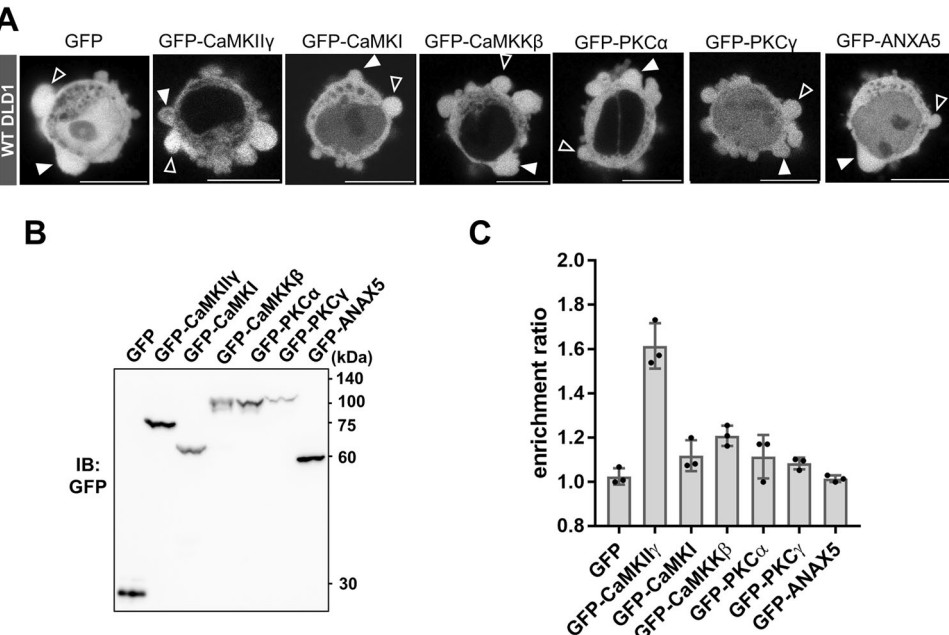

**Figure EV1.  Screening for cytoplasmic Ca²⁺-related proteins based on their enrichment within blebs.**

(**A**) Wild-type DLD1 cells expressing GFP-tagged cytoplasmic proteins subjected to screening. Black arrowheads indicate expanding blebs, and white arrowheads indicate retracting blebs (Scale bar: 10 μm). (**B**) Western blot analysis of HEK293 cell lysates expressing the vectors used for screening, probed with an anti-GFP antibody. (**C**) Maximum enrichment ratio of each GFP-tagged protein relative to monomeric RFP within expanding blebs. Quantification was performed for three blebs per protein and results are shown as mean of three biological replicates ± SD.

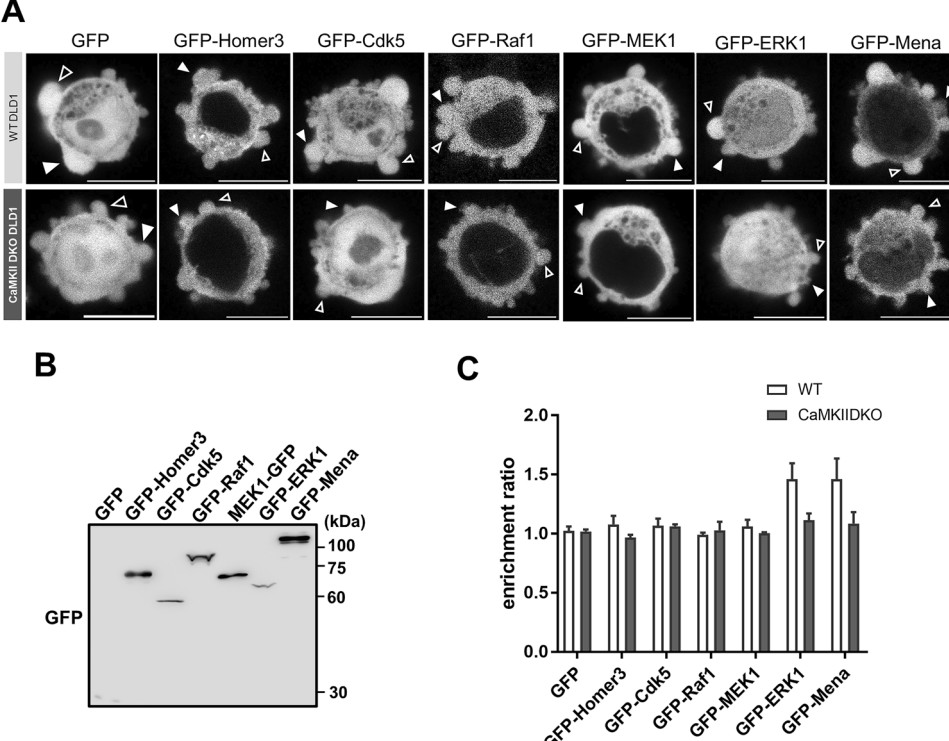

**Figure EV2.  Analysis of bleb accumulation of GFP-tagged CaMKII-related cytoplasmic proteins.**

(**A**) Wild-type DLD1 cells expressing GFP-tagged cytoplasmic proteins reported to interact with CaMKII. Black arrowheads indicate expanding blebs, and white arrowheads indicate retracting blebs (Scale bar: 10 µm). (**B**) Western blot analysis of HEK293 cell lysates expressing the vectors used for localization analysis, probed with an anti-GFP antibody. (**C**) Quantification of the maximum enrichment ratio of each GFP-tagged protein relative to monomeric RFP within expanding blebs. Five blebs were analyzed for each protein and results are shown as mean of three biological replicates ± SD.

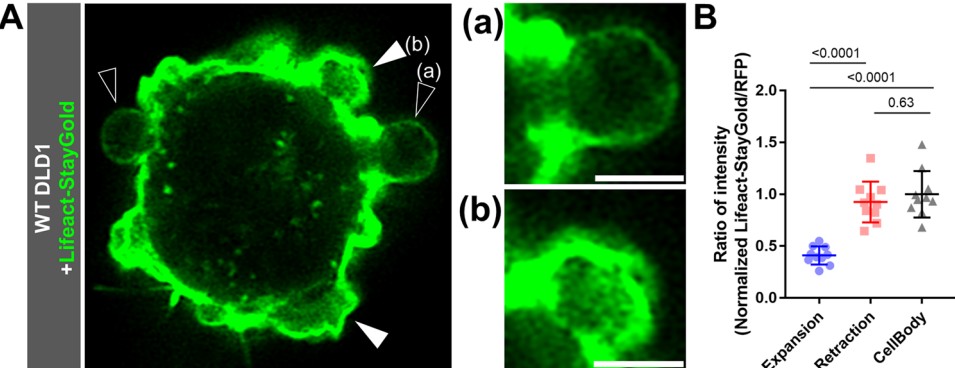

**Figure EV3.  Super-resolution imaging of the cytoplasmic actin network during bleb expansion and retraction.**

(A) Super-resolution images of WT cells expressing Lifeact-StayGold, acquired with Airyscan2 and processed by Airyscan processing. Black arrowheads indicate expanding blebs, while white arrowheads indicate retracting blebs (Scale bar: 10 μm). Panels show (a) an expanding bleb, (b) a retracting bleb (Scale bar: 2 μm). Notably, differences in actin organization are observed not only at the cortex but also within the cytoplasm. (B) WT cells co-expressing Lifeact-StayGold and RFP were imaged using Airyscan2, and super-resolution images obtained after Airyscan processing were used for analysis. For each bleb, the mean fluorescence intensities of Lifeact-StayGold and RFP were measured in three cytoplasmic regions: the interior of expanding blebs, the interior of retracting blebs, and the cell-body cytoplasm adjacent to the bleb. The ratio of Lifeact-StayGold to RFP intensity was calculated for each region. Quantification was performed on 10 blebs, followed by one-way ANOVA with Tukey's multiple comparisons test; $p$ values are $p < 0.0001$ (for Expansion vs. Retraction), $p < 0.0001$ (for Expansion vs. CellBody), $p = 0.6293$ (for Retraction vs. CellBody).

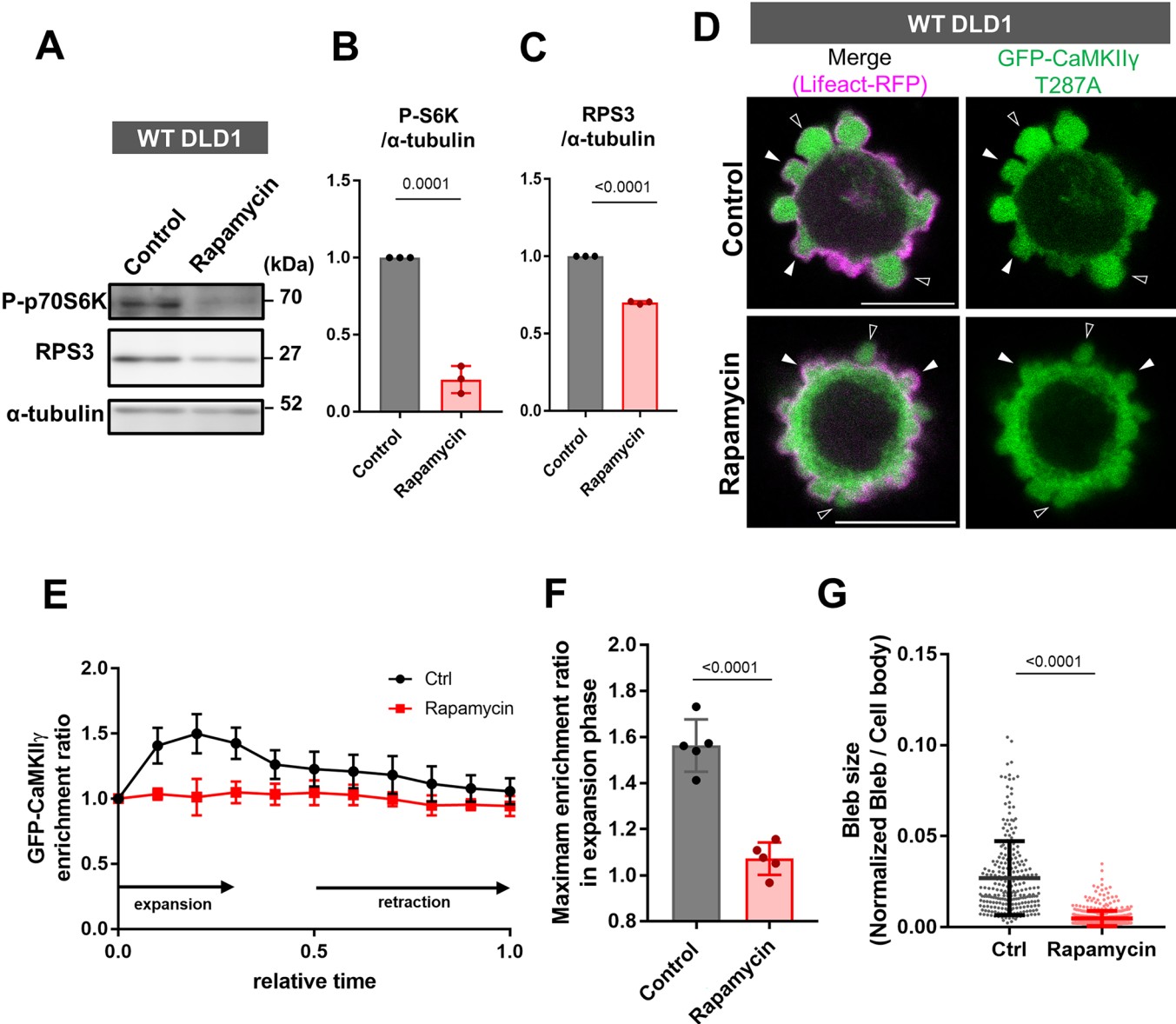

**Figure EV4.  Effects of rapamycin-induced ribosome depletion on CaMKII dynamics and bleb morphology.**

(A) Western blot analysis of lysates from wild-type DLD1 cells treated with the indicated reagents. α-Tubulin was used as a loading control. (B) Quantification of the Western blot results. A marked decrease in phospho-p70S6K levels was observed in rapamycin-treated cells, confirming effective inhibition of mTORC1. Data represents the mean ± SD from three independent experiments. Statistical significance was Student's t-test; $p$ value is $p = 0.0001$. (C) Quantification of the Western blot results. Rapamycin-treated cells showed a significant reduction in RPS3 signal intensity. Each experiment was performed three times. Data represents the mean ± SD from three independent experiments. Statistical significance was Student's t-test; $p$ value is $p < 0.0001$. (D) Fluorescence microscopy images of DLD1 cells expressing GFP–CaMKII and Lifeact–RFP after treatment with the indicated reagents. Black arrowheads indicate expanding blebs, while white arrowheads indicate retracting blebs. In cells treated with rapamycin, accumulation of GFP–CaMKII within expanding blebs was markedly reduced, accompanied by a decrease in bleb size. (E) Temporal changes in CaMKII enrichment levels within blebs of cells treated with various reagents. Each dataset represents quantification of five blebs, and the mean ± SD is plotted for each time point. (F) Maximum enrichment ratio of GFP–CaMKII relative to monomeric RFP within expanding blebs of cells treated with the indicated reagents. Quantification was performed for five blebs per group, and statistical significance was assessed using Student's t-test; $p$ value is $p < 0.0001$. (G) Distribution of bleb sizes in WT cells treated with indicated agents. The areas of all blebs in 20 cells were quantified. Results are shown as mean of four biological replicates ± SD and the $p$ values of Student's *t*-test are indicated; $p$ value is $p < 0.0001$.

