## [Peer Review File · The EMBO Journal]

CaMKII nucleates an osmotic protein supercomplex to induce cellular bleb expansion

Yuki Fujii, Yuji Sakai, Kenji Matsuzawa, and Junichi Ikenouchi

Corresponding author: Junichi Ikenouchi (ikenouchi.junichi.033@m.kyushu-u.ac.jp)

Review Timeline:

Submission Date:	11th Jun 25
Editorial Decision:	31st Jul 25
Revision Received:	29th Nov 25
Editorial Decision:	2nd Jan 26
Revision Received:	4th Jan 26
Accepted:	9th Jan 26

Editor: Ieva Gailite

Transaction Report:

Dear Dr. Ikenouchi,

Thank you for submitting your manuscript for consideration by the EMBO Journal. We have now received comments from a full set of reviewers, which are included below for your information.

As you will see, all reviewers express interest in the findings and appreciate their novelty. At the same time, they also find that further in-depth analysis to support the proposed model would be needed before they can support publication here. In particular, the role of osmotic pressure and the causal role of CAMKII vs actin for bleb formation would need further verification. During the reviewer cross-commenting session, clarification of the time-course of CMAKII accumulation in comparison to actin meshwork formation was highlighted as an experiment that could provide further support to the proposed model. Furthermore, all reviewers indicate that the kinase activity and properties regarding bleb induction of the used CAMKII mutants will need to be further characterised. Finally, they also request to provide more information on the experimental approach and the screen leading to CAMKII identification, and to provide better statistical analysis of the data. Experimental analysis of CAMKII condensate formation in the current context as mentioned by reviewer #3 as non-essential will not be required for acceptance here, but can be well included in the discussion for a broader context.

Based on these generally positive assessments, I invite you to submit a revised manuscript in response to the comments by all reviewers. I think that it would be useful to discuss the revision in more detail via email or phone/videoconferencing. However, please note that I will be away from the office until August 18 and will be able to arrange a direct Zoom call only after my return.

We generally allow three months as standard revision time, which can be extended to six months in the case of major revisions. Should you foresee a problem in meeting this deadline, please let us know in advance to discuss an extension.

As a matter of policy, competing manuscripts published during this period will not negatively impact on our assessment of the conceptual advance presented by your study. However, please contact me as soon as possible upon publication of any related work to discuss the appropriate course of action.

When preparing your letter of response to the referees' comments, please bear in mind that this will form part of the Review Process File and will therefore be available online to the community. For more details on our Transparent Editorial Process, please visit our website: <https://www.embopress.org/page/journal/14602075/authorguide#transparentprocess>. Please also see the attached instructions for further guidelines on preparation of the revised manuscript.

Please feel free to contact me if you have any further questions regarding the revision. Thank you for the opportunity to consider your work for publication. I look forward to discussing your revision.

With best wishes,

Ieva

We realize that it is difficult to revise to a specific deadline. In the interest of protecting the conceptual advance provided by the work, we recommend a revision within 3 months (29th Oct 2025). Please discuss the revision progress ahead of this time with the editor if you require more time to complete the revisions.

Referee #1:

The paper by Fujii et al. aims to decipher the mechanisms that facilitate bleb expansion. Specifically, the authors examine the role of CaMKII γ , a molecule that functions at the region where the bleb forms. Indeed, the corresponding knockout led to a different morphology/size of cellular blebs.

The authors suggest that CaMKII, a kinase, plays a novel role in bleb formation, with the surprising finding that the role is independent of the kinase activity.

Rather, the authors suggest a function in controlling osmotic pressure that induces bleb expansion. Thus, the claim is that CaMKII functions not only as a kinase but also as a scaffold regulating local osmotic pressure and membrane deformation.

If correct, the finding is interesting and worth publishing in the Journal. Nevertheless, the reservations listed below question the conclusions and model that the authors put forward. The main issue that is key for conclusions is the independence of kinase activity, as presented in the specific points, in particular point 13.

Comments are in order of presentation in the text.

1. Unless relying on experiments conducted in this work, the authors should cite primary research references that back up specific, highly relevant statements they make.
e.g. "During bleb formation, a pressure difference arises between the cell body, where the contractile actin cortex maintains high pressure, and the bleb, which lacks this cortex and thus exhibits lower pressure.", or "Interestingly, intracellular Ca²⁺ level is elevated in the expanding blebs..."
2. "Using GFP-tagged Ca²⁺-associated proteins in human colon cancer DLD1 cells, we performed live-cell imaging to screen for factors that accumulate in expanding blebs.- The authors should describe the screen in the results section. Which proteins were checked, and what were the results of the screen?"
3. Many conclusions rely upon GFP activity. It would be good if at least a few main findings were also backed up by anti-GFP antibody staining. The activity/fluorescence of specific fusions might be enhanced at specific parts of the cells (e.g. blebs) that differ in pH, osmolarity etc.
4. In Figure 1E, the bleb volume is presented in circles while the key on the graph shows squares.
5. For panel 1F, provide the range of time one observes - what is the range of 100% time
6. "We first generated CaMKII-deficient DLD1 cells to investigate their contribution to bleb dynamics"- the cells were generated to study the potential role of CaMKII in bleb dynamics, not the cells.

7. In Figure 2B,C, the bleb size is decreased, but the number of blebs appears higher and is not quantified. Please quantify. If indeed the number is increased, what is the underlying mechanism?
8. In the graph in Figure 2B, the average bleb size of the DKO cells appears about 2 times larger than that of the DKO cells presented in Figure 2D, which raises questions regarding the reliability of the measurements/quantifications. The transfection of the DKO cells with the tagged CaMKIIy has a minor effect on the bleb size (Fig 2D), but the effect appears very strong in Fig 2E, so the example presented in 2E does not represent the average. This does not fit the statement "since the expression of CRISPR-resistant CaMKIIy in CaMKII DKO cells restored bleb size to WT levels (Figs. 2D and 2E)", since the average bleb size in WT cells is 0.05 in fig 2B, but less than 0.02 in the CaMKIIy-expressing CaMKII DKO cells.
9. In Fig 3C, it seems like the inactivated version of the protein (T287A) enhances blebbing to a level higher than that observed in knockout cells expressing the wildtype/activity-reduced mutant. What is the explanation for that?
10. Given the issue of very large size of blebs in the T287A mutant as compared with the wild type, it is not clear why use it as a control - "We therefore opted to use the T287A mutant as a control in subsequent experiments since it allowed us to exclude autophosphorylation-dependent activity change as a confounding factor in interpreting any results.". Based on this sentence, this mutant is considered a wild type, although it shows a phenotype.
11. In Figure 5D, the triple mutant shows less blebbing than the knockout. What is the explanation for that?
12. Since the authors have the relevant GFP fusion constructs, it would help if the expression levels of the different protein versions were measured to show equal levels and stability.
13. Given the "provocative message", the effects the mutations have on kinase activity should be demonstrated in the cells the authors use, to ensure that it is indeed completely abolished in the relevant context. Another protein version that should be included in the study would be a mutation/s that eliminate the kinase activity directly (e.g. ATP binding), rather than mutations that only regulate it.
- 14 In movie 1, the open CaMKII conformation appears to represent a result of the expanded bleb rather than having any causative entity. e.g. the bleb forming at the right at 1:05 expands even though "open configuration" is found internally, and the highest level of the open state is observed only after the full expansion.

Referee #2:

This manuscript presents an intriguing mechanism by which the ability of CaMKII to unfold and bind to calmodulin and other downstream substrates drives water flow into blebs to increase their maximum size before the bleb retracts into the plasma membrane. While the proposed mechanism is quite novel and unexpected, the evidence supporting it is not strong and rests on many untested assumptions. Further, the authors rely on extensive discussion of their findings in the context of the published literature within the results section to introduce some of their approaches, rather than relying on the internal logic of the experimental data to tell the story. Many of the figures lack adequate labels, making it unclear exactly what was being tested or analyzed in several instances. Overall, the data does not always support the stated conclusions, and this results in the authors overinterpreting their findings in several key instances, reducing enthusiasm for the manuscript in its current form.

Comments:

General

The labels on the figures are not sufficient to convey what the figures are showing. For example, the figures don't always indicate the cell type (e.g. double ko) or what was being measured (e.g. GFP-mena concentration ratio), what is being pulled down versus blotted, which mutants are being imaged and analyzed. Strongly suggest adding more labels so the general experimental details can be inferred from just looking at the figure rather than relying on figure legends or descriptions in the text.

More description of the DLD1 cells and how they were cultured is required in the paper and materials and methods. The cells all appear rounded up and non-adherent. How were the cells cultured to prevent adhesion to the underlying substrate? Was there any experimental manipulation performed to induce the observed blebbing?

The appropriate statistical tests need to be performed on all the data in order to support the conclusions. For example, no statistical tests are evident in Figure 6B and F.

Figure 1

The first paragraph of the results presents itself as a recapitulation of their previous paper. This is very confusing, and it is unclear what information is new and why it is necessary to confirm the experiments from the previous paper? It seems more straightforward to shift the first paragraph of the results to the introduction and cite the findings of the previous paper. And then begin the paper with a novel experiment (the localization of CaMKII).

I strongly suggest the authors include some of the negative results from their screen for GFP-tagged calcium-associated proteins that identified CamKII. It would be very informative to include some negative results to highlight the novelty of CamKII.

The structure and function of the proteins that did not accumulate in the expanding blebs or did not rescue their maximum size could demonstrate what is unique about CaMKII in this process.

The authors need to confirm that endogenous CamKII is capable of becoming enriched in expanding blebs as they see for the overexpressed GFP-tagged versions.

Figure 3

Why is the GFP control in Figure 3B missing from quantification presented in Figure 3D? This should be included.

Figure 4

The Camui-CR FRET probe requires positive and negative controls to confirm that it is reporting the folded state of CaMKII. The quantification should be based on a whole cell measurement since the FRET signal should not depend on where in the cell it is located in the case of an inhibitor treatment like KN93.

The results discussion Figure 4B mention the differences in the FRET signal in expanding blebs, collapsing blebs, and within the cytoplasm. Representative time lapse images with each of these different locations clearly labelled need to be included. Further, the differences in FRET signal must be measured with the appropriate statistical tests performed in order to support the concept that the FRET signal depends on where the FRET probed is localized within the cell.

Given that KN93 and EDTA treatment prevent the enrichment of CaMKII in the expanding blebs and reduce their maximum size, the importance of measuring the folded state of CaMKII in the blebs under these conditions is unclear. Perhaps it would be more helpful to compare the other regions of the cell, to show that the cytoplasmic CaMKII is maintained in a folded state under those conditions?

Figure 5

Figure 5 would benefit from including a diagram of CaMKII illustrating the location/size of the mutants used in this figure.

Figure 6 and S1

The labels on these figures are not sufficient, doesn't always indicated cell type (e.g. double ko) or what was being measured (e.g. GFP-mena concentration ratio), what is being pulled down versus blotted. Strongly suggest adding more labels so the general experimental details can be inferred from just looking at the figure rather than relying on figure legends or descriptions in the text. This applies to all of the figures in the paper, not just figure 6.

Figure 6D needs to be quantified.

I strongly suggest the authors include some of the negative results from their screen for GFP-tagged CaMKII binding partners. It would again be very informative to include some negative results to highlight the novelty of Mena. The structure and function of the proteins that did not accumulate in the expanding blebs could help demonstrate what is unique about Mena in this process.

The authors claim that a fine actin meshwork is what enables CaMKII to accumulate in blebs with almost no evidence. This needs to be experimentally verified if this claim is going to remain in the paper. The images presented in S1 are simply not convincing. Some sort of FRAP based experiments with drugs targeting F-actin assembly, disassembly could be get some much stronger evidence to support this proposed mechanism.

Similarly, the claims about the presence of CaMKII in the blebs as changing the osmotic pressure to drive inward flow of water are very flimsy. This again needs to be tested directly, perhaps by varying the osmolarity of the media and measuring the rate of bleb expansion in the presence or absence of CaMKII. Osmotic pressure is mainly governed by number of particles, rather than size of particles, not clear why smaller particles could enter or leave the blebs via the F-actin meshwork to change the overall osmotic pressure.

The fact that FsaA-YPet also accumulates in blebs and can rescue the loss CaMKII is very puzzling. If any large protein can compensate for CaMKII, why would it matter if CaMKII was compact, elongated, alone, or in a larger complex? Raises concerns about doing these experiments with overexpressed proteins, it may be the amount of protein expressed is far more important than other aspects of its structure and function. The amount of GFP-tagged proteins in the different cells being analyzed may need to be controlled for.

Figure 7

The images presented in Figure 7A, B, and C need to be quantified as earlier in the manuscript and appropriate statistical tests

performed. For the FRET probe, comparing the signal from the bleb versus cytoplasm would also strengthen the data.

For C and D, since this is a new cell type, the authors must confirm that KN93 treatment reduces bleb size and dynamics. Further that CaMKII knockdown slows cell movement and also reduces bleb size and dynamics and that CaMKII overexpression can rescue these features and cell motility. Without these confirming experiments, the conclusion that CaMKII governs bleb size and dynamics to enable low-adhesive confined migration is very tenuous.

Referee #3:

This study by Fuji et al convincingly demonstrates a non-enzymatic function of the ubiquitous CaMKII gamma and/or delta isozymes in membrane blebbing. The approach exceeds the recently proposed standards for studying CaMKII (Brown and Bayer 2024, Cell Reports): Even though only a single CaMKII inhibitor is used, the confirmation by molecular genetic approaches is even more convincing than a back up by a second inhibitor class. The genetic approach specifically included knockout and re-expression of different mutants, including the ATP-binding incompetent K43M mutant that abolishes any enzymatic activity. Notably, if this mutant had not restored WT function, the mechanisms could still be either enzymatic or structural (as abolished ATP-binding not only prevents enzymatic activity but also reduces binding to some proteins, such as GluN2B). However, as the K43M mutant restored WT function, it unequivocally demonstrates that enzymatic kinase activity is not required for this function. Thus, in my opinion, no additional experiments are required for the current study, and any mentioning of experiments in my comments below are solely suggestions for possible future follow-up studies. As such, the revision should require only some re-writing but no additional experiments. My most notable suggestion is regarding additional discussion and comparison to the LLPS that has been prominently described for CaMKII and has received much attention. The mechanism described here appears to be very similar and to constitute some form of biomolecular condensation. However, while the function of LLPS with CaMKII in neurons is still entirely speculative (with some emerging evidence even suggesting that some of these speculations are incorrect), the authors here clearly demonstrate a function of their mechanism. In my opinion, this is a strong point of the current study that is "under-sold" in the current version.

"major" suggestion:

1) The CaMKII mechanism described here bears strong similarity to the CaMKII co-condensation with GluN2B that has been prominently proposed to mediate synaptic organization and plasticity. Thus, the discussion should include a more extensive comparison. In addition to ref 17 (Hosokawa et al 2021 Nature Neuroscience), this should include Cai et al (2021) Cell Res 31:37-51. For instance, maintenance of the co-condensates beyond the a Ca²⁺ stimulus required autophosphorylation of T286 (=T287 in non-alpha isozymes); consequently, persistence of the co-condensates (but not their induction) was abolished by the T286A and K42M mutation. By contrast, in the current study, the corresponding mutations did not ablate the blebbing functions. This should be discussed as it appears to be a discrepancy, but really is not (as it can be easily explained by the persistent Ca²⁺ elevation in the blebs that make mechanisms to outlast such Ca²⁺ stimuli unnecessary).

Much beyond the scope of the current study (but as suggestion for possible future follow-ups), this comparison between the mechanisms could be extended experimentally. For instance, one expectation might be that chelation of Ca²⁺ in the blebs by EGTA-AM might collapse the blebs in DKO cell that is re-expressing T287A or K43M mutant CaMKIIgamma, but not when WT is re-expressed instead. Additionally, one would expect that re-expression of truncated CaMKII monomers that do not form co-condensates (Hosokawa et al 2021) would not support blebbing, but that full-length CaMKII monomers (with mutated hub domain) might support blebbing, as these were just this week published to support co-condensation (Brown et al 2025 bioRxiv "CaMKII monomers are sufficient for GluN2B binding, co-condensation, and synaptic potentiation").

One additional comparison to the co-condensation of CaMKII with GluN2B at synapses could be pointed out in favor of the current manuscript: While functions of this co-condensation are intriguing but still entirely speculative, the mechanism described in the current manuscript has a demonstrated function in blebbing. For co-condensation, a role in synaptic plasticity (specifically LTP) has been proposed. (See the original papers by Hosokawa and by Cai; as well as subsequent review articles such as Chen et al 2020 Nat Neurosci or Liu et al 2021 Curr Opin Neurobiol). However, directly supporting experimental evidence is lacking; and any existing experimental evidence even suggests that at least the co-condensation with GluN2B has not role in LTP (bioRxiv manuscript above and Rumian et al 2024 Cell Rep). The authors may choose not to emphasize this point, as without such explicit emphasis, it remains a major strength of the current study.

Minor comments:

2) P5 L10: the referenced review for "memory encoding" provides an outdated model indicating that pT286 maintains the memory, however, a slew of newer data indicates that this is not the case, with definite evidence provided over the last two years. As pT286 still is involved in generating memory (even if not storing it), the sentence can be kept largely the same, but needs to cite the newer evidence, for instance as reviewed in Bayer and Giese (2025) Nature Neuroscience.

3) P6 L21: the wording regarding "GluN2B acting as the glue securing CaMKII dodecamers" is misleading. While ref 17 suggests that CaMKII dodecamers are required for the co-condensation with GluN2B, the dodecamers are stable without such co-condensation. Thus, the sentence needs to be reworded. Also, while ref 17 is an appropriate reference for CaMKII co-

condensation with GluN2B, the second reference should not be current ref 18 but Cai et al (2021) Cell Res 31:37-51.

4) P8 L2 and P10 L20: The "open conformation" is a bit misleading, especially in connection with the illustration in Fig. 4A (which should also be changed). It suggests that the extended holoenzyme conformation is the active one; by contrast, it is the activation competent form. Note that the cited ref 21 shows that the extended disk-like structure is formed even without Ca²⁺/CaM. Without Ca²⁺/CaM, these extended holoenzymes are inactive but activation-competent. Only a minor fraction of kinase subunits is in the activation-incompetent compact conformation. This observation in ref 21 is also consistent with cryo-EM studies by the Stratton lab [Sloutsky et al (2020) Sci Signal] and single-molecule atomic force microscopy by the Shibata lab [Tsujioka et al (2023) Sci Adv].

5) P10 L16 and Fig 6: Recruitment of erk1 (in addition to mena) is interesting. It would be interesting to test in the future if blebbing is affected by erk inhibitors. (In case this is already known, it should be mentioned and cited). Minor note: A direct interaction has been described between CaMKII and raf1, an upstream activating kinase for erk1.

6) P9-P14: As this results sub-section is rather long (and only half deals with the "osmotic activity"), it might be beneficial to separate this into multiple subsections.

7) Fig. 1A: While EDTA can chelate Ca²⁺ ions, it favors chelating Mg²⁺ over Ca²⁺. Thus, the effect may be generated by chelation of Mg²⁺ ions (which would indeed be expected to affect the cell membrane) rather than Ca²⁺ ions. This could be easily alleviated by instead using EGTA, which allows effective chelation of Ca²⁺ but still substantial free Mg²⁺. Additionally, intracellular chelators (EGTA-AM or BAPTA-AM) may verify the role of intracellular Ca²⁺. While it is not necessary to repeat these experiments (based on the demonstrated involvement of SOCE for Ca²⁺ influx and the clear demonstration of requirement for stimulation of CaMKII by Ca²⁺/CaM), there should be some discussion of EDTA vs EGTA and acknowledgement that EDTA primarily chelates Mg²⁺ (so to not mislead others). In this context, it is important to disclose the concentration of EDTA used here, as well as the concentrations of Mg²⁺ and Ca²⁺ in the medium.

8) It would be helpful if the authors could provide estimates of the Ca²⁺ concentrations within the blebs in the discussion. However, this is not essential, as they clearly demonstrate a significant increase over basal Ca²⁺ in the cell body.

9) It should be specifically disclosed which form of GFP was used for fusion with CaMKII. I assume at least EGFP; and possibly even mEGFP with the further monomerizing mutation. Either way, this should be stated in the methods.

10) In Fig. 6A, change "CaMKII" to "CaMKII DKO" in the image label.

11) It might be helpful to mention the cell type in every Fig legend (as three different cell types used overall).

12) The famous aminoacid T286 of CaMKII is correctly identified as T287 in the other isoforms. However, similarly, K42M in CaMKII alpha is K43M in the other isoforms. This should be corrected.

Referee #1

The paper by Fujii et al. aims to decipher the mechanisms that facilitate bleb expansion. Specifically, the authors examine the role of CaMKII γ , a molecule that functions at the region where the bleb forms. Indeed, the corresponding knockout led to a different morphology/size of cellular blebs.

The authors suggest that CaMKII, a kinase, plays a novel role in bleb formation, with the surprising finding that the role is independent of the kinase activity.

Rather, the authors suggest a function in controlling osmotic pressure that induces bleb expansion. Thus, the claim is that CaMKII functions not only as a kinase but also as a scaffold regulating local osmotic pressure and membrane deformation.

If correct, the finding is interesting and worth publishing in the Journal. Nevertheless, the reservations listed below question the conclusions and model that the authors put forward. The main issue that is key for conclusions is the independence of kinase activity, as presented in the specific points, in particular point 13. Comments are in order of presentation in the text.

Comment 1-1

1. Unless relying on experiments conducted in this work, the authors should cite primary research references that back up specific, highly relevant statements they make.

e.g. "During bleb formation, a pressure difference arises between the cell body, where the contractile actin cortex maintains high pressure, and the bleb, which lacks this cortex and thus exhibits lower pressure.", or "Interestingly, intracellular Ca²⁺ level is elevated in the expanding blebs..."

Response 1-1

Thank you for this helpful comment. In response, we carefully reviewed the relevant primary literature and have now added appropriate citations for the statements in question.

Comment 1-2

"Using GFP-tagged Ca²⁺-associated proteins in human colon cancer DLD1 cells, we performed live-cell imaging to screen for factors that accumulate in expanding blebs.- The authors should describe the screen in the results section. Which proteins were checked, and what were the results of the screen?"

Response 1-2

As suggested, we have now provided a detailed description of the live-cell imaging screen in the Results section. We examined the localization dynamics of several GFP-tagged cytoplasmic proteins regulated by Ca²⁺ in wild-type DLD1 cells, including CaMKI, CaMKII, CaMKK β , PKC α , PKC γ , and ANXA5, during bleb expansion. Among these candidates, only CaMKII showed clear

enrichment in expanding blebs. The other proteins did not accumulate in blebs or were evenly distributed between the blebs and the cell body. We have summarized these results in **Figure EV1** and have added a detailed description of the screen to the Results section.

Comment 1-3

Many conclusions rely upon GFP activity. It would be good if at least a few main findings were also backed up by anti-GFP antibody staining. The activity/fluorescence of specific fusions might be enhanced at specific parts of the cells (e.g. blebs) that differ in pH, osmolarity etc.

Response 1-3

Thank you for raising this important point. We fully agree that GFP fluorescence can be influenced by local pH, osmolarity, or other environmental factors, and that such effects may cause apparent enrichment at specific cellular sites (e.g., blebs) that do not necessarily reflect true protein accumulation.

To verify that the increased GFP-CaMKII fluorescence intensity observed inside expanding blebs reflects actual protein accumulation, we fixed DLD1 cells expressing GFP-CaMKII and performed immunofluorescence staining using an anti-GFP antibody (**Appendix Fig. S1**). The results confirmed that GFP-CaMKII was indeed enriched within expanding blebs, supporting that the observed localization represents genuine protein accumulation rather than local enhancement of GFP fluorescence.

Comment 1-4

In Figure 1E, the bleb volume is presented in circles while the key on the graph shows squares.

Response 1-4

Thank you for pointing this out. We have corrected the legend to display circles, consistent with the data. The revised figure has been updated in the manuscript accordingly.

Comment 1-5

For panel 1F, provide the range of time one observes - what is the range of 100% time

Response 1-5

In Fig. 1E (previously Fig. 1F), “100%” in the relative time axis represents the time from the initiation of bleb formation, through expansion, until the completion of bleb retraction. To avoid overcrowding the figure, we did not include temporal changes in bleb volume in panels after Fig. 1E.

However, for all figures describing time-dependent changes, the relative time was defined based on the lifetime of each analyzed bleb (i.e., the duration from bleb initiation to completion of retraction).

The actual lifetimes of individual blebs varied, typically ranging from approximately 60 to 90 seconds. By normalizing time in this way, we found that nearly all blebs reached maximal expansion at 20–40% of the relative time and underwent retraction after 50%. Therefore, we adopted the relative time scale to facilitate visualization and evaluation of bleb dynamics, particularly during the expansion phase, which is difficult to compare statistically using absolute time values.

The detailed analytical procedure and the definition of relative time are described in the Quantification of Protein Enrichment in the Cytoplasm of Expanding Blebs section of the Methods.

Comment 1-6

"We first generated CaMKII-deficient DLD1 cells to investigate their contribution to bleb dynamics"- the cells were generated to study the potential role of CaMKII in bleb dynamics, not the cells.

Response 1-6

Thank you for pointing this out. We have revised the sentence as follows:

"To investigate the potential role of CaMKII in bleb dynamics, we generated CaMKII-deficient DLD1 cells."

This revision has been incorporated into the main text.

Comment 1-7

In Figure 2B,C, the bleb size is decreased, but the number of blebs appears higher and is not quantified. Please quantify. If indeed the number is increased, what is the underlying mechanism?

Response 1-7

We thank the reviewer for this insightful comment. We have quantified the total cell volume, the volume of blebs, and the number of blebs per cell for both wild-type and CaMKII DKO cells. The analysis revealed that the number of blebs per cell was significantly increased in CaMKII DKO cells compared with wild-type cells (wild-type: 59.33 ± 4.22 ; CaMKII DKO: 101.50 ± 20.83).

Although we have not experimentally verified the underlying mechanism, we would like to offer a possible interpretation. Intracellular pressure is generated as the contractile actomyosin cortex compresses the cytoplasm, producing hydrostatic pressure that pushes the plasma membrane

outward. Under steady-state conditions, cortical tension and hydrostatic pressure are balanced. Previous studies have shown that in highly rounded cells with elevated cortical tension, bleb expansion leads to a significant decrease in intracellular pressure, likely because a portion of the cytoplasmic volume contributing to the pressure is displaced into the expanding bleb. These studies also suggest the existence of a threshold level of hydrostatic pressure required for bleb formation and growth (for example, by local laser ablation of the cortex in rounded cells; see Tinevez et al., *PNAS* 2009).

Since there were no significant differences in total cell volume or myosin phosphorylation levels between wild-type and CaMKII DKO cells, the overall hydrostatic pressure across the cell is likely comparable. We speculate that, in wild-type cells where CaMKII accumulates in expanding blebs and promotes their growth, a single bleb can rapidly relieve local pressure once the critical threshold for bleb formation is exceeded. In contrast, CaMKII DKO cells form only small blebs that are insufficient to relieve hydrostatic pressure, thereby maintaining a high-pressure state that may continuously trigger the formation of new blebs, resulting in an increased number of blebs overall.

Because this proposed mechanism remains speculative and is not directly related to the main conclusions of our study, we have kept the discussion in the main text to a minimum.

Comment 1-8

In the graph in Figure 2B, the average bleb size of the DKO cells appears about 2 times larger than that of the DKO cells presented in Figure 2D, which raises questions regarding the reliability of the measurements/quantifications. The transfection of the DKO cells with the tagged CaMKII γ has a minor effect on the bleb size (Fig 2D), but the effect appears very strong in Fig 2E, so the example presented in 2E does not represent the average. This does not fit the statement "since the expression of CRISPR-resistant CaMKII γ in CaMKII DKO cells restored bleb size to WT levels (Figs. 2D and 2E)", since the average bleb size in WT cells is 0.05 in fig 2B, but less than 0.02 in the CaMKII γ -expressing CaMKII DKO cells.

Response 1-8

Thank you for your comment. In the previous version of the manuscript, the experiments shown in Figures 2B and 2D were conducted at different time points, comparing WT and DKO cells in one experiment, and DKO and Rescue cells in another. To address this concern, we have now performed a new set of experiments in which WT, DKO, and Rescue (CaMKII γ re-expression in DKO) cells were analyzed side by side under strictly controlled and identical conditions. These experiments were independently repeated three times using the same serum lot and synchronized cell handling to

minimize variability. The results of these experiments are now presented in **Figure 2C**.

Consistent with our previous observations, we confirmed that the average bleb size was significantly reduced in CaMKII DKO cells compared to wild-type cells. Importantly, re-expression of CRISPR-resistant CaMKII γ in DKO cells restored bleb size to levels comparable to those observed in wild-type cells.

Comment 1-9

In Fig 3C, it seems like the inactivated version of the protein (T287A) enhances blebbing to a level higher than that observed in knockout cells expressing the wildtype/activity-reduced mutant. What is the explanation for that?

Response 1-9

We thank the reviewer for this insightful comment. Although the blebs in cells expressing the T287A mutant in Figure 3D appear slightly larger, our quantitative analysis demonstrated no significant differences in average bleb size among CaMKII DKO cells rescued with wild-type CaMKII γ , the T287A mutant, or the T287A/K43M double mutant. These data are presented in **Figure 3E**.

Comment 1-10

Given the issue of very large size of blebs in the T287A mutant as compared with the wild type, it is not clear why use it as a control - "We therefore opted to use the T287A mutant as a control in subsequent experiments since it allowed us to exclude autophosphorylation-dependent activity change as a confounding factor in interpreting any results.". Based on this sentence, this mutant is considered a wild type, although it shows a phenotype.

Response 1-10

We thank the reviewer for raising this important point. As mentioned in our response to Comment 1-9, we re-evaluated bleb size using fixed samples under strictly controlled conditions (identical culturing and imaging parameters), and performed quantitative analysis across CaMKII DKO cells rescued with wild-type CaMKII γ , the T287A mutant, and the T287A/K43M double mutant. This analysis revealed no statistically significant differences in bleb size among these conditions (see Fig. 3E).

We used the T287A mutant as a control in subsequent experiments for the following reason. As noted in the main text, autophosphorylation at T287 induces a conformational change that maintains CaMKII in a persistently active state, independent of Ca²⁺/calmodulin (CaM). Since our goal was to

dissect the contribution of Ca²⁺-dependent CaM binding to CaMKII function, the T287A mutant, which is unable to undergo autophosphorylation and thus lacking this Ca²⁺-independent conformational stabilization, served as an appropriate control to isolate the Ca²⁺/CaM-dependent CaMKII activity. Therefore, despite its mutation, T287A provides a well-defined functional baseline for interpreting downstream effects in a Ca²⁺-dependent context.

Comment 1-11

In Figure 5D, the triple mutant shows less blebbing than the knockout. What is the explanation for that?

Response 1-11

We thank the reviewer for this valuable comment. In response, we repeated the quantification using fixed samples under matched experimental conditions (including cell culture and imaging settings) for CaMKII DKO cells and those rescued with T287A, T287A/I205K, and T287A/T305D/T306D mutants. These additional analyses, now presented in the revised **Figure 5F**, revealed no statistically significant differences in bleb size among these conditions. In particular, the T287A/T305D/T306D triple mutant did not show a consistent trend toward reduced blebbing compared to DKO cells.

Comment 1-12

Since the authors have the relevant GFP fusion constructs, it would help if the expression levels of the different protein versions were measured to show equal levels and stability.

Response 1-12

We thank the reviewer for this important suggestion. For the stable expression of GFP-tagged CaMKII γ variants used in the quantification of bleb size and kinase activity in CaMKII double-knockout DLD1 cells, we employed FACS to isolate populations with comparable GFP expression levels. Following sorting, we performed Western blot analyses to confirm the molecular weight, expression levels, and protein stability of each construct. These analyses demonstrated that all GFP-CaMKII variants were expressed at comparable levels, matched the expected molecular weights, and showed no signs of degradation. We also verified the molecular weight and stability of the GFP-tagged proteins used in the calcium-dependent screening assays by Western blotting. These validations are presented in the following figures:

GFP-CaMKII γ rescue constructs (GFP, WT CaMKII, T287A, K42M): **Figure 3B**

Additional rescue constructs (T287A, Δ C, 2TD, I205K): **Figure 5D**

Calcium-related screening (GFP-tagged proteins): **Figure EV1B**

CaMKII-related screening (GFP-tagged proteins): **Figure EV2B**

These data demonstrate the equivalent expression and stability of the different protein variants used in our study.

Comment 1-13

Given the "provocative message", the effects the mutations have on kinase activity should be demonstrated in the cells the authors use, to ensure that it is indeed completely abolished in the relevant context. Another protein version that should be included in the study would be a mutation/s that eliminate the kinase activity directly (e.g. ATP binding), rather than mutations that only regulate it.

Response 1-13

We thank the reviewer for this important point. As noted, previous studies have demonstrated that the T287A mutation specifically disrupts the autonomous activation of CaMKII by preventing autophosphorylation at Thr287, while the K43M inactivates the kinase function by disrupting the ATP-binding pocket within the catalytic domain (Hanson et al. *Neuron* 1994; Rich and Schulman *J Biol Chem* 1998).

Given that one of the central messages of our study is that CaMKII contributes to bleb formation in a manner independent of its kinase activity, we performed direct kinase activity measurements using CaMKII DKO DLD1 cells stably expressing different GFP-CaMKII variants. Specifically, we utilized the CycLex CaM-kinase II Assay Kit (MBL) to quantify CaMKII enzymatic activity in lysates prepared from these cells.

Our results show that while cells rescued with wild-type CaMKII displayed robust kinase activity, both the T287A and the T287A/K42M double mutant exhibited only minimal activity, comparable to cells expressing GFP alone as a negative control. These findings confirm that these mutants are catalytically inactive under our experimental conditions and validate our interpretation that the observed bleb-related phenotypes occur independently of CaMKII kinase activity.

We have added a summary of these results in the revised manuscript and referenced the corresponding data in **Figure 3C**.

Comment 1-14

In movie 1, the open CaMKII conformation appears to represent a result of the expanded bleb rather than having any causative entity. e.g. the bleb forming at the right at 1:05 expands even though

"open configuration" is found internally, and the highest level of the open state is observed only after the full expansion.

Response 1-14

We appreciate the reviewer's thoughtful observation regarding the temporal relationship between the open conformation of CaMKII and bleb expansion in Movie EV1. As noted, the DLD1 cells used in our study exhibit dynamic blebbing around the entire cell periphery. However, since the movie represents a single confocal plane, it is possible that a bleb appearing to have reached its maximal expansion in the focal plane may still be actively expanding in a different z-plane. This optical limitation can occasionally lead to an apparent mismatch between bleb expansion and CaMKII conformational changes.

Nevertheless, the overall pattern consistently shows that the conformational change of CaMKII occurs during bleb expansion, rather than after full expansion. Importantly, this pattern is distinct from the general cytosolic distribution of CaMKII, as shown in **Figure 4E**. Furthermore, we provide quantitative evidence in **Figures 4F and 4G**, demonstrating that FRET efficiency significantly decreases in synchrony with bleb enlargement. These data support our interpretation that the open conformation of CaMKII is closely associated with the expansion phase of blebs.

Referee #2

This manuscript presents an intriguing mechanism by which the ability of CaMKII to unfold and bind to calmodulin and other downstream substrates drives water flow into blebs to increase their maximum size before the bleb retracts into the plasma membrane. While the proposed mechanism is quite novel and unexpected, the evidence supporting it is not strong and rests on many untested assumptions. Further, the authors rely on extensive discussion of their findings in the context of the published literature within the results section to introduce some of their approaches, rather than relying on the internal logic of the experimental data to tell the story. Many of the figures lack adequate labels, making it unclear exactly what was being tested or analyzed in several instances. Overall, the data does not always support the stated conclusions, and this results in the authors overinterpreting their findings in several key instances, reducing enthusiasm for the manuscript in its current form.

Comment 2-1

The labels on the figures are not sufficient to convey what the figures are showing. For example, the figures don't always indicate the cell type (e.g. double ko) or what was being measured (e.g.

GFP-mena concentration ratio), what is being pulled down versus blotted, which mutants are being imaged and analyzed. Strongly suggest adding more labels so the general experimental details can be inferred from just looking at the figure rather than relying on figure legends or descriptions in the text.

Response 2-1

We appreciate the reviewer's suggestion. In response, we have revised the figures to include clearer labels indicating the cell type, measured parameters, IP/WB antibody combinations, and CaMKII variants used in each experiment.

Comment 2-2

More description of the DLD1 cells and how they were cultured is required in the paper and materials and methods. The cells all appear rounded up and non-adherent. How were the cells cultured to prevent adhesion to the underlying substrate? Was there any experimental manipulation performed to induce the observed blebbing?

Response 2-2

We appreciate the reviewer's comment. As noted, DLD1 cells were seeded onto non-coated glass-bottom dishes, and we observed that they spontaneously formed stable blebs approximately 6 hours after seeding, without any additional treatment. No chemical stimulation or manipulation was applied to induce blebbing. We have added this information to the Materials and Methods section of the revised manuscript.

Comment 2-3

The appropriate statistical tests need to be performed on all the data in order to support the conclusions. For example, no statistical tests are evident in Figure 6B and F.

Response 2-3

We appreciate the reviewer's comment regarding statistical analyses. We have now performed appropriate quantitative analyses for all figures. For the data shown in the previous Figure 6B and 6F (now Figure 6H in the revised manuscript), we conducted statistical tests comparing the degree of enrichment of GFP-Mena or GFP-ERK at each time point during bleb expansion and retraction between wild-type and CaMKII DKO cells. In addition, we have included new quantitative data in **Figure 6C** and **Figure 6I** showing the peak enrichment of GFP-Mena or GFP-ERK at blebs during the expansion phase relative to RFP, and statistically compared these values between the two cell types.

Comment 2-4

The first paragraph of the results presents itself as a recapitulation of their previous paper. This is very confusing, and it is unclear what information is new and why it is necessary to confirm the experiments from the previous paper? It seems more straightforward to shift the first paragraph of the results to the introduction and cite the findings of the previous paper. And then begin the paper with a novel experiment (the localization of CaMKII).

Response 2-4

We appreciate the reviewer's comment and fully agree that Figure 1A–F summarizes our previous findings showing Ca²⁺ accumulation in expanding blebs, as reported by Aoki et al. *Nat Commun* 2021. We intentionally included these results as a minimal background because this Ca²⁺ enrichment during bleb expansion is not yet widely recognized, particularly among readers who are more familiar with CaMKII signaling but less acquainted with bleb biology. We have clarified this rationale in the revised text and slightly condensed the first paragraph of the Results to make the transition to our new CaMKII-focused findings (from Figure 1G onward) clearer.

Comment 2-5

I strongly suggest the authors include some of the negative results from their screen for GFP-tagged calcium-associated proteins that identified CamKII. It would be very informative to include some negative results to highlight the novelty of CamKII. The structure and function of the proteins that did not accumulate in the expanding blebs or did not rescue their maximum size could demonstrate what is unique about CaMKII in this process.

Response 2-5

We thank the reviewer for this helpful suggestion. This point overlaps with Reviewer #1's Comment 1-2, to which we have already responded by adding a detailed description of the live-cell imaging screen in the Results section. Specifically, we now describe that several GFP-tagged cytoplasmic Ca²⁺-regulated proteins (including CaMKI, CaMKII, CaMKK β , PKC α , PKC γ , and ANXA5) were examined, and that only CaMKII showed clear enrichment in expanding blebs, while the others did not. These results are summarized in **Figure EV1** and described in detail in the revised Results section.

Comment 2-6

The authors need to confirm that endogenous CamKII is capable of becoming enriched in expanding blebs as they see for the overexpressed GFP-tagged versions.

Response 2-6

We appreciate the reviewer's suggestion to verify the localization of endogenous CaMKII. In response, we performed immunofluorescence staining of DLD1 cells during bleb formation and confirmed that endogenous CaMKII was enriched within expanding blebs. These results have been added to the revised manuscript as **Figure 1J**.

Comment 2-7

Why is the GFP control in Figure 3B missing from quantification presented in Figure 3D? This should be included.

Response 2-7

We thank the reviewer for pointing out this omission. As suggested, we have now included the negative control data for CaMKII DKO cells expressing GFP alone in the quantification shown in **Figure 3D** (previously Figure 3B).

Comment 2-8

Figure 4. The Camui-CR FRET probe requires positive and negative controls to confirm that it is reporting the folded state of CaMKII. The quantification should be based on a whole cell measurement since the FRET signal should not depend on where in the cell it is located in the case of an inhibitor treatment like KN93.

Response 2-8

We appreciate the reviewer's comment regarding the validation of the Camui-CR FRET probe. To confirm that Camui-CR functions properly in DLD1 cells, we expressed the probe in fully adherent DLD1 cells that had been cultured for 24 hours (a non-blebbing condition) and monitored FRET efficiency before and after pharmacological treatments. Ionomycin treatment decreased FRET efficiency, whereas EDTA and KN93 treatments increased FRET efficiency, consistent with the expected conformational changes of CaMKII. The quantification was based on whole-cell measurements rather than local regions. These results are shown in **Figure 4C** of the revised manuscript.

Comment 2-9

The results discussion Figure 4B mention the differences in the FRET signal in expanding blebs, collapsing blebs, and within the cytoplasm. Representative time lapse images with each of these different locations clearly labelled need to be included. Further, the differences in FRET signal must be measured with the appropriate statistical tests performed in order to support the concept that the

FRET signal depends on where the FRET probed is localized within the cell.

Response 2-9

We appreciate the reviewer's comment. To demonstrate that the decrease in FRET efficiency occurs specifically within expanding blebs, we included time-lapse images showing both the bleb and the adjacent cytoplasmic region at the same time points (**Figure 4E**). We also quantified the changes in FRET efficiency in the bleb region and the neighboring cytoplasmic region during bleb formation (**Figures 4F and 4G**).

Comment 2-10

Given that KN93 and EDTA treatment prevent the enrichment of CaMKII in the expanding blebs and reduce their maximum size, the importance of measuring the folded state of CaMKII in the blebs under these conditions is unclear. Perhaps it would be more helpful to compare the other regions of the cell, to show that the cytoplasmic CaMKII is maintained in a folded state under those conditions?

Response 2-10

We appreciate the reviewer's insightful comment. Although KN93 and EDTA treatment reduce bleb size, small blebs are still formed under these conditions. To test whether CaMKII might exhibit structural differences between blebs and the surrounding cytoplasm independent of Ca²⁺/CaM activation, we analyzed FRET efficiency using the Camui-CR probe. As expected, both treatments strongly inhibited Ca²⁺/CaM-dependent activation of CaMKII throughout the cytoplasm, resulting in uniformly high FRET efficiency without any regional difference. Because this finding essentially confirms the global inhibition of CaMKII activity, we have simplified these data and moved them to the Supplementary Information (now **Appendix Figure S2**). The main text now refers to this observation briefly.

Comment 2-11

Figure 5 would benefit from including a diagram of CaMKII illustrating the location/size of the mutants used in this figure.

Response 2-11

We appreciate the reviewer's suggestion to include a schematic illustration to clarify the structural states and localization of CaMKII mutants. In response, we have created a new diagram in **Figure 5E** summarizing the presumed conformational states and subcellular localization of each CaMKII mutant during bleb formation, together with the corresponding bleb size.

Comment 2-12

Figure 6 and S1. The labels on these figures are not sufficient, doesn't always indicated cell type (e.g. double ko) or what was being measured (e.g. GFP-mena concentration ratio), what is being pulled down versus blotted. Strongly suggest adding more labels so the general experimental details can be inferred from just looking at the figure rather than relying on figure legends or descriptions in the text. This applies to all of the figures in the paper, not just figure 6.

Response 2-12

We appreciate the reviewer's valuable suggestion. In the revised version, we have added detailed labels not only to Figure 6 but also to all figure panels throughout the manuscript, so that the general experimental design and conditions can be understood directly from the figures without referring to the legends or main text.

Comment 2-13

Figure 6D needs to be quantified.

Response 2-13

In relation to Figure 6D (now Figure 6E in the revised manuscript), we analyzed the localization of Mena in CaMKII DKO cells rescued with various Scarlet-CaMKII mutants. The fluorescence intensities of Mena in expanding blebs and in the cell body regions were quantified, and the results are shown in **Figure 6F** demonstrating how each CaMKII mutant differentially affects Mena accumulation within blebs.

Comment 2-14

I strongly suggest the authors include some of the negative results from their screen for GFP-tagged CaMKII binding partners. It would again be very informative to include some negative results to highlight the novelty of Mena. The structure and function of the proteins that did not accumulate in the expanding blebs could help demonstrate what is unique about Mena in this process.

Response 2-14

We appreciate the reviewer's suggestion to include the results of our screen for GFP-tagged CaMKII-binding partners. We have now added these data as **Figure EV2**. In this screen, we examined several cytoplasmic proteins endogenously expressed in DLD1 cells that have been reported to interact with CaMKII, together with Mena, which was previously shown to accumulate within expanding blebs (Aoki et al. *Nat Commun* 2021). The results revealed that, in wild-type DLD1 cells, both Mena and ERK1 were prominently enriched within expanding blebs, whereas in

CaMKII DKO cells, neither protein accumulated in the bleb interior.

Comment 2-15

The authors claim that a fine actin meshwork is what enables CaMKII to accumulate in blebs with almost no evidence. This needs to be experimentally verified if this claim is going to remain in the paper. The images presented in S1 are simply not convincing. Some sort of FRAP based experiments with drugs targeting F-actin assembly, disassembly could be get some much stronger evidence to support this proposed mechanism.

Response 2-15

We appreciate the reviewer's insightful comment regarding the need for experimental verification of our proposed mechanism that a fine actin meshwork enables CaMKII accumulation within blebs. As suggested, we attempted to perform FRAP-based experiments under pharmacological perturbation of F-actin dynamics using latrunculin B (LatB). Treatment with micromolar concentrations (5-10 μM) of LatB suppressed cortical actin recovery and bleb retraction in under a minute and lower concentrations down to 0.5 μM simply delayed this effect. There was no notable change in bleb expansion/retraction dynamics at even lower concentrations down to 0.1 μM , suggesting that the selectively disrupting the cytoplasmic actin meshwork while preserving cortical actin formation—and thus normal bleb dynamics—is technically unfeasible by pharmacological perturbation. That is, there is no LatB concentration range that selectively disassembles cortical or cytoplasmic actin without globally impairing cell morphology. Therefore, FRAP experiments under actin-disrupting conditions could not be used to evaluate the contribution of the actin meshwork to CaMKII accumulation within blebs.

Mechanistically, we consider that CaMKII accumulating within expanding blebs exists predominantly in its Ca^{2+} -dependent, open conformation, which forms multivalent interactions with other proteins. These interactions likely reduce its molecular mobility and make it less able to penetrate into the structurally constrained cytoplasmic regions of the cell body. In contrast, CaMKII in the cell body remains largely in an inactive, compact form with fewer intermolecular interactions and only modest restriction by the cytoplasmic solid phase, which we discuss below. Thus, although the diffusion of CaMKII is suppressed in both compartments, the physical mechanisms differ: intermolecular binding within the bleb versus steric hindrance in the cell body. The interface between these two regions, each imposing distinct types of diffusional constraints, likely acts synergistically to restrict transboundary diffusion, resulting in a pronounced accumulation of CaMKII within expanding blebs.

For this reason, even if FRAP were performed in LatB-treated cells to compare CaMKII diffusivity within the cytoplasm, such experiments would primarily reflect how inactive CaMKII is constrained by the cytoplasmic solid phase, rather than revealing the synergistic diffusional restriction and accumulation mechanism that occurs between the expanding bleb and the adjacent cell body.

In summary, while we fully agree with the reviewer on the importance of verifying the mechanism experimentally, the technical limitations of actin perturbation and the complex interplay of diffusion constraints between distinct cytoplasmic domains preclude a straightforward FRAP-based validation at present.

It is well established that the cytoplasm forms a porous, viscoelastic gel phase (“cytoplasmic solid phase”) organized by an actin meshwork in cooperation with macromolecules and organelles such as ribosomes (Etoc et al. *Nat Mater* 2018). Previous study reported that reduction of ribosome content decreases the density of this cytoplasmic solid phase and consequently accelerates mesoscale (tens of nanometers) molecular diffusion (Delarue et al *Cell* 2018). Based on this, we hypothesized that modulating the cytoplasmic solid phase would affect CaMKII accumulation in blebs.

We therefore treated DLD1 cells with 1 μ M rapamycin for 3 h, a condition known to reduce ribosome abundance, and examined GFP-CaMKII localization. Under these conditions, the enrichment of CaMKII within expanding blebs was lost, and the blebs themselves became smaller. This result indicates that the cytoplasmic solid phase, supported by the actin meshwork, plays an important role in CaMKII accumulation within blebs (**Figure EV4**).

Previously we have shown that the diffusivity of quantum dots is higher in expanding blebs than in retracting blebs or the cell body, and comparable to that in the cytoplasm of cells treated with actin polymerization inhibitors (Aoki et al. *Nat Commun* 2021). Together with electron microscopy data showing the presence of actin meshwork in the cytoplasm and retracting blebs (Chikina et al. *J Cell Biol.* 2019), these findings strongly suggest distinct physical states of the cytoplasmic solid phase between expanding blebs and other regions. The novelty of our **Figure EV3** (previous Figure S1) lies in the application of live-cell super-resolution microscopy to visualize the cytoplasmic actin meshwork in blebbing cells. Because the cytoplasmic actin meshwork signal is weak and highly dynamic, conventional probes and imaging methods were insufficient to reliably capture its structure. To overcome this limitation, we employed the exceptionally photostable Lifact-StayGold probe in combination with high-speed super-resolution imaging, which enabled us to visualize the fine actin meshwork in both expanding and retracting blebs. The resulting images clearly show that expanding blebs exhibit a marked reduction of cytoplasmic actin meshwork, whereas retracting blebs retain a

denser network, consistent with previous ultrastructural observations. To further support these observations, we quantified the ratio of Lifeact-StayGold to co-expressed monomeric RFP (**Figure EV3B**), which revealed a significant decrease specifically during bleb expansion, confirming that the reduced relative signal reflects a genuine loss of the cytoplasmic actin meshwork.

Collectively, although direct pharmacological dissection was technically limited, our rapamycin-based approach supports the idea that CaMKII accumulation depends on the integrity of the cytoplasmic solid phase by reducing cytoplasmic ribosome content and thereby loosening the cytoplasmic solid phase. Together with our imaging data showing the presence of the cytoplasmic actin meshwork in cell body, these results suggest that the structural integrity of cytoplasmic solid phase is critical for CaMKII enrichment.

Comment 2-16

Similarly, the claims about the presence of CaMKII in the blebs as changing the osmotic pressure to drive inward flow of water are very flimsy. This again needs to be tested directly, perhaps by varying the osmolarity of the media and measuring the rate of bleb expansion in the presence or absence of CaMKII. Osmotic pressure is mainly governed by number of particles, rather than size of particles, not clear why smaller particles could enter or leave the blebs via the F-actin meshwork to change the overall osmotic pressure.

Response 2-16

We thank the reviewer for this insightful comment. We believe that part of the concern may arise from a misunderstanding of how osmotic pressure contributes to bleb expansion in our model. We do not propose that bleb growth is driven by water influx from the extracellular space. This view is supported by the work of Raz and colleagues (Goudarzi et al., *PLOS ONE*, 2019), who performed high-resolution imaging of zebrafish primordial germ cells and showed that overall cell volume changes only minimally during bleb expansion and that no detectable water enters the bleb from the extracellular space. They concluded that the major source of volume for bleb expansion is cytoplasmic redistribution rather than extracellular influx. Our model is consistent with this framework.

To directly address the reviewer's suggestion, we performed additional experiments manipulating extracellular osmolarity under both hyperosmotic and hypoosmotic conditions and analyzed their effect on bleb formation. Consistent with Charras et al. (*Nature*, 2005), hyperosmotic treatment reduced intracellular hydrostatic pressure and led to bleb shrinkage in both WT and CaMKII DKO cells (**Figures R1A and R1B**). In contrast, hypoosmotic treatment increased cell volume and

internal pressure (**Figures R1C and R1D**), and under these conditions we observed clear phenotypic differences between CaMKII DKO and CaMKII-rescue cells: CaMKII-rescue cells exhibited pronounced bleb enlargement, whereas CaMKII-DKO cells showed limited bleb expansion and instead displayed swelling of the cell body (**Figures R1E and R1F**).

	CaMKII DKO	CaMKII-rescue
Number of Blebs	Before : 10.67 ± 2.52 After : 17.67 ± 1.53	Before : 5.67 ± 2.51 After : 8.67 ± 2.08
Bleb size (Bleb/Cell body)	Before : 0.011 ± 0.0076 After : 0.011 ± 0.0055	Before : 0.062 ± 0.041 After : 0.072 ± 0.048

Table: Temporal analysis of Bleb dynamics during the 2 minutes Pre- and Post-Hypo-osmotic Treatment

Figure R1 Morphological changes of blebs in cells under hyperosmotic or hypoosmotic conditions

(A) Phase-contrast images of WT and CaMKII DKO cells treated with hyperosmotic medium (600 mOsm).

Scale bar: 10 μ m.

(B) The rate of volume change in WT and CaMKII DKO cells treated with hyperosmotic medium. Volumes

were inferred from the area measurement. The areas of whole cells, three cells per cell type, were quantified before and after treatment. Data represent the means \pm SD. The p-value from Student's t-test is shown.

- (C) Phase-contrast images of CaMKII DKO and CaMKII-rescue cells treated with hypoosmotic medium (200 mOsm). Scale bar: 10 μ m.
- (D) The rate of volume change in CaMKII DKO and CaMKII-rescue cells treated with hypoosmotic medium. The areas of whole cells, three cells per cell type, were quantified before and after treatment. Data represent the means \pm SD. The p-value from Student's t-test is shown (right). The effects of hypotonic stimulation on cell and bleb volume was immediate but was attenuated within a few minutes due to the cell's ability to regulate their volume. For this reason, the sizes of cells and blebs were quantified by live imaging rather than in fixed cells as in the experiments included in the main text.
- (E) The volumes of cell bodies in three cells per cell type before and after treatment with hypoosmotic medium were quantified. Data represent the means \pm SD. The p-value from Student's t-test is shown.
- (F) Distribution of bleb sizes in CaMKII DKO and CaMKII-rescue cells before and after treatment with hypoosmotic medium. The bleb areas were measured for all blebs formed within a two minute period immediately before and after hypoosmotic treatment using live image data. The area at maximum bleb expansion was measured for each bleb. Three cells were analyzed for each cell type. The p-value from one-way ANOVA followed by Tukey's post hoc test is shown. The data is summarized in the table below.

These findings align well with our model in which supramolecular assemblies formed by CaMKII exert osmotic activity that increases the relative osmolarity within expanding blebs, thereby attracting water from the cytoplasm and enabling these blebs to accommodate internal pressure. In CaMKII-rescue cells, this activity allows individual blebs to take up sufficient water and expand. In CaMKII-DKO cells, the absence of CaMKII-dependent osmotic activity prevents adequate water accumulation inside blebs, reducing their expansion capacity. Consequently, when hypoosmotic stress increases whole-cell water content, DKO cells cannot dissipate pressure through bleb growth and instead expand at the level of the cell body.

In addition to these new osmotic perturbation experiments, the main manuscript already includes quantitative analyses directly relevant to this issue. As shown in Appendix Table S1, we quantified the concentrations of CaMKII, Mena, and ERK in blebs versus the cell body, revealing substantial molecular concentration gradients. Based on these experimentally determined protein concentrations, the resulting osmotic pressure differences were further calculated using a mathematical model (Page 15–16), demonstrating that their magnitudes are sufficient to drive membrane deformation. Moreover, the markedly smaller blebs observed in CaMKII-DKO cells (**Figure 2B**) further support

the conclusion that CaMKII-dependent osmotic activity contributes to bleb expansion.

It is also important to consider that osmotic stress affects not only intracellular water content and hydrostatic pressure but also other mechanical properties of the cell. Moeendarbary et al. (*Nat. Mater.* 2013) showed that changes in cell volume alter cytoplasmic pore size through poroelastic mechanisms, thereby influencing the movement of water molecules and soluble proteins. Because changes in extracellular osmolarity simultaneously impact multiple mechanical parameters, they cannot be used to directly infer the specific contribution of intracellular osmotic gradients. For these reasons, a comprehensive quantitative model describing how osmotic perturbations reshape bleb morphology will require future investigation.

Comment 2-17

The fact that FsaA-YPet also accumulates in blebs and can rescue the loss CaMKII is very puzzling. If any large protein can compensate for CaMKII, why would it matter if CaMKII was compact, elongated, alone, or in a larger complex? Raises concerns about doing these experiments with overexpressed proteins, it may be the amount of protein expressed is far more important than other aspects of its structure and function. The amount of GFP-tagged proteins in the different cells being analyzed may need to be controlled for.

Response 2-17

We appreciate the reviewer's thoughtful comment regarding the accumulation of FsaA-YPet in blebs and its ability to rescue the loss of CaMKII. FsaA is an *E. coli*-derived, heterologous protein that can exist in two distinct states, a monomeric form and a stable decamer. In expanding blebs, monomeric FsaA likely enters the bleb interior and then assembles into large decamers that are too bulky to pass through the actin meshwork, resulting in its accumulation inside blebs. Although this behavior is intriguing, it reflects an artificial phenomenon arising from the expression of a non-native bacterial protein. Because elucidating the molecular basis of FsaA accumulation is beyond the scope of this study, we have removed the related data from the revised manuscript.

As for the reviewer's concern about protein expression levels, we confirmed by Western blotting that the expression levels of CaMKII mutants were comparable among samples, as shown in **Figures 3B and 5D**. Therefore, the observed differences in localization and bleb morphology are not attributable to differences in protein abundance.

Comment 2-18

The images presented in Figure 7A, B, and C need to be quantified as earlier in the manuscript and

appropriate statistical tests performed. For the FRET probe, comparing the signal from the bleb versus cytoplasm would also strengthen the data.

Response 2-18

We thank the reviewer for this helpful suggestion. As requested, we have performed quantitative analyses and statistical evaluations for the data shown in **Figures 7B, 7D, and 7E**. Specifically, we quantified the fluorescence intensity of the GCaMP signal and the degree of CaMKII accumulation in blebs, using the same analytical approach and statistical tests as applied in the earlier figures. The quantified results have been added to the revised figure panels and legends.

Comment 2-19

Figure 7. For C and D, since this is a new cell type, the authors must confirm that KN93 treatment reduces bleb size and dynamics. Further that CaMKII knockdown slows cell movement and also reduces bleb size and dynamics and that CaMKII overexpression can rescue these features and cell motility. Without these confirming experiments, the conclusion that CaMKII governs bleb size and dynamics to enable low-adhesive confined migration is very tenuous.

Response 2-19

We thank the reviewer for this valuable comment. As suggested, we examined the effect of CaMKII inhibition on bleb formation in Walker 256 cells. We quantified bleb size in control and KN-93–treated cells and found that inhibition of CaMKII resulted in the formation of significantly smaller blebs, indicating that CaMKII activity contributes to bleb expansion in this cell type (**Figures 7F and 7G**). These results are consistent with our observations in DLD1 cells.

Walker 256 cells predominantly express the CaMKII γ isoform. However, these cells exhibit extremely low transfection efficiency and are resistant to lentiviral infection. We attempted to introduce CRISPR-Cas9 by electroporation to knock out CaMKII γ , but were unable to obtain stable knockout clones. Therefore, a detailed phenotypic analysis of CaMKII γ -deficient Walker 256 cells remains a subject for future investigation.

Referee #3

This study by Fuji et al convincingly demonstrates a non-enzymatic function of the ubiquitous CaMKII gamma and/or delta isozymes in membrane blebbing. The approach exceeds the recently proposed standards for studying CaMKII (Brown and Bayer 2024, Cell Reports): Even though only a single CaMKII inhibitor is used, the confirmation by molecular genetic approaches is even more convincing than a back up by a second inhibitor class. The genetic approach specifically included

knockout and re-expression of different mutants, including the ATP-binding incompetent K43M mutant that abolishes any enzymatic activity. Notably, if this mutant had not restored WT function, the mechanisms could still be either enzymatic or structural (as abolished ATP-binding not only prevents enzymatic activity but also reduces binding to some proteins, such as GluN2B). However, as the K43M mutant restored WT function, it unequivocally demonstrates that enzymatic kinase activity is not required for this function. Thus, in my opinion, no additional experiments are required for the current study, and any mentioning of experiments in my comments below are solely suggestions for possible future follow-up studies. As such, the revision should require only some re-writing but no additional experiments. My most notable suggestion is regarding additional discussion and comparison to the LLPS that has been prominently described for CaMKII and has received much attention. The mechanism described here appears to be very similar and to constitute some form of biomolecular condensation. However, while the function of LLPS with CaMKII in neurons is still entirely speculative (with some emerging evidence even suggesting that some of these speculations are incorrect), the authors here clearly demonstrate a function of their mechanism. In my opinion, this is a strong point of the current study that is "under-sold" in the current version.

Comment 3-1

The CaMKII mechanism described here bears strong similarity to the CaMKII co-condensation with GluN2B that has been prominently proposed to mediate synaptic organization and plasticity. Thus, the discussion should include a more extensive comparison. In addition to ref 17 (Hosokawa et al 2021 Nature Neuroscience), this should include Cai et al (2021) Cell Res 31:37-51. For instance, maintenance of the co-condensates beyond the a Ca²⁺ stimulus required autophosphorylation of T286 (=T287 in non-alpha isozymes); consequently, persistence of the co-condensates (but not their induction) was abolished by the T286A and K42M mutation. By contrast, in the current study, the corresponding mutations did not ablate the blebbing functions. This should be discussed as it appears to be a discrepancy, but really is not (as it can be easily explained by the persistent Ca²⁺ elevation in the blebs that make mechanisms to outlast such Ca²⁺ stimuli unnecessary).

Much beyond the scope of the current study (but as suggestion for possible future follow-ups), this comparison between the mechanisms could be extended experimentally. For instance, one expectation might be that chelation of Ca²⁺ in the blebs by EGTA-AM might collapse the blebs in DKO cell that is re-expressing T287A or K43M mutant CaMKII γ , but not when WT is re-expressed instead. Additionally, one would expect that re-expression of truncated CaMKII monomers that do not form co-condensates (Hosokawa et al 2021) would not support blebbing, but that full-length CaMKII monomers (with mutated hub domain) might support blebbing, as these were just this week published to support co-condensation (Brown et al 2025 bioRxiv "CaMKII monomers are sufficient for GluN2B binding, co-condensation, and synaptic potentiation").

One additional comparison to the co-condensation of CaMKII with GluN2B at synapses could be pointed out in favor of the current manuscript: While functions of this co-condensation are intriguing but still entirely speculative, the mechanism described in the current manuscript has a demonstrated function in blebbing. For co-condensation, a role in synaptic plasticity (specifically LTP) has been proposed. (See the original papers by Hosokawa and by Cai; as well as subsequent review articles such as Chen et al 2020 *Nat Neurosci* or Liu et al 2021 *Curr Opin Neurobiol*). However, directly supporting experimental evidence is lacking; and any existing experimental evidence even suggests that at least the co-condensation with GluN2B has not role in LTP (bioRxiv manuscript above and Rumian et al 2024 *Cell Rep*). The authors may choose not to emphasize this point, as without such explicit emphasis, it remains a major strength of the current study.

Response 3-1

We thank the reviewer for this insightful suggestion. We have now added a new paragraph in the Discussion section comparing our findings with the CaMKII–GluN2B condensates previously described in neurons (Hosokawa et al., 2021; Cai et al., 2021; Brown and Bayer, 2024). This addition highlights both the similarities, such as Ca^{2+} /CaM-dependent supramolecular assembly, and the key differences arising from distinct Ca^{2+} dynamics. Specifically, in expanding blebs, sustained cytoplasmic Ca^{2+} elevation maintains CaMKII in an activation-competent conformation without the need for autophosphorylation at T287, which is required to maintain neuronal condensates after transient Ca^{2+} spikes. We have also emphasized that, in contrast to neuronal LLPS whose physiological role remains speculative, our study demonstrates a defined mechanical function of CaMKII-based condensates in driving membrane deformation.

Comment 3-2

P5 L10: the referenced review for "memory encoding" provides an outdated model indicating that pT286 maintains the memory, however, a slew of newer data indicates that this is not the case, with definite evidence provided over the last two years. As pT286 still is involved in generating memory (even if not storing it), the sentence can be kept largely the same, but needs to cite the newer evidence, for instance as reviewed in Bayer and Giese (2025) *Nature Neuroscience*.

Response 3-2

We agree and have updated the reference to include the recent review by Bayer and Giese (*Nat Neurosci* 2025), which provides an updated perspective on the role of pT286/pT287 autophosphorylation in encoding, but not storing, synaptic plasticity. The corresponding sentence has been revised accordingly.

Comment 3-3

P6 L21: the wording regarding "GluN2B acting as the glue securing CaMKII dodecamers" is misleading. While ref 17 suggests that CaMKII dodecamers are required for the co-condensation with GluN2B, the dodecamers are stable without such co-condensation. Thus, the sentence needs to be reworded. Also, while ref 17 is an appropriate reference for CaMKII co-condensation with GluN2B, the second reference should not be current ref 18 but Cai et al (2021) *Cell Res* 31:37-51.

Response 3-3

We thank the reviewer for pointing this out. We have revised the sentence to clarify that GluN2B promotes multivalent cross-linking and co-condensation between CaMKII holoenzymes, rather than stabilizing the dodecamers themselves. We also added a citation to Cai et al. (*Cell Res* 2021). The revised sentence now reads:

Page 7 Line 17

“Intriguingly, interaction with GluN2B, itself a tetramer, not only stabilizes the active state of CaMKII but also promotes its liquid–liquid phase separation (LLPS) by mediating multivalent cross-linking between CaMKII dodecamers (Hosokawa et al., 2021; Cai et al., 2021).”

Comment 3-4

P8 L2 and P10 L20: The "open conformation" is a bit misleading, especially in connection with the illustration in Fig. 4A (which should also be changed). It suggests that the extended holoenzyme conformation is the active one; by contrast, it is the activation competent form. Note that the cited ref 21 shows that the extended disk-like structure is formed even without Ca²⁺/CaM. Without Ca²⁺/CaM, these extended holoenzymes are inactive but activation-competent. Only a minor fraction of kinase subunits is in the activation-incompetent compact conformation. This observation in ref 21 is also consistent with cryo-EM studies by the Stratton lab [Sloutsky et al (2020) *Sci Signal*] and single-molecule atomic force microscopy by the Shibata lab [Tsujioka et al (2023) *Sci Adv*].

Response 3-4

We agree and have replaced “open conformation” with “activation-competent conformation” throughout the manuscript. We have also added citations to Sloutsky et al. (*Sci. Signal.*, 2020) and Tsujioka et al. (*Sci. Adv.*, 2023), which demonstrate that CaMKII holoenzymes adopt extended activation-competent conformations upon Ca²⁺/CaM binding.

Comment 3-5

P10 L16 and Fig 6: Recruitment of erk1 (in addition to mena) is interesting. It would be interesting to test in the future if blebbing is affected by erk inhibitors. (In case this is already known, it should be mentioned and cited). Minor note: A direct interaction has been described between CaMKII and raf1, an upstream activating kinase for erk1.

Response 3-5

We appreciate the reviewer's thoughtful comment and suggestion to consider the link between CaMKII and the Raf–MAPK pathway. We tested whether upstream components of the ERK pathway, such as Raf-1 and MEK, are recruited to blebs, but found that both proteins remained cytoplasmic and did not accumulate in blebs. This was also the case in CaMKII DKO cells (**Figure EV2**). These observations suggest that the association of CaMKII with ERK1 in blebs is unlikely to be mediated through Raf-1 or MEK.

We have revised the text to clarify this point and to avoid implying that ERK1 recruitment depends on canonical Raf–MEK signaling. We still acknowledge that CaMKII can interact with Raf-1 in other cellular contexts (Illario et al., 2003; Salzano et al., 2012), but our data indicate that such an interaction does not occur at blebs.

We agree with the reviewer that it will be interesting in the future to test whether inhibition of ERK activity influences bleb dynamics, as this could provide further insight into how CaMKII-dependent signaling contributes to membrane behavior.

Comment 3-6

P9-P14: As this result sub-section is rather long (and only half deals with the "osmotic activity"), it might be beneficial to separate this into multiple subsections.

Response 3-6

We agree and have divided the original section describing osmotic activity into two distinct subsections for improved clarity:

- (1) "CaMKII conformational change and supramolecular assembly in expanding blebs" and
- (2) "Osmotic contribution of CaMKII-based complexes to bleb growth (CODE: CaMKII-based osmotically-driven deformation; CODE)"

This structural change separates the molecular findings from the biophysical modeling, improving readability and logical flow.

Comment 3-7

Fig. 1A: While EDTA can chelate Ca^{2+} ions, it favors chelating Mg^{2+} over Ca^{2+} . Thus, the effect may be generated by chelation of Mg^{2+} ions (which would indeed be expected to affect the cell membrane) rather than Ca^{2+} ions. This could be easily alleviated by instead using EGTA, which allows effective chelation of Ca^{2+} but still substantial free Mg^{2+} . Additionally, intracellular chelators (EGTA-AM or BAPTA-AM) may verify the role of intracellular Ca^{2+} . While it is not necessary to repeat these experiments (based on the demonstrated involvement of SOCE for Ca^{2+} influx and the clear demonstration of requirement for stimulation of CaMKII by Ca^{2+} /CaM), there should be some discussion of EDTA vs EGTA and acknowledgement that EDTA primarily chelates Mg^{2+} (so to not mislead others). In this context, it is important to disclose the concentration of EDTA used here, as well as the concentrations of Mg^{2+} and Ca^{2+} in the medium.

Response 3-7

We thank the reviewer for this important point. We have now specified the EDTA concentration (5 mM) and the Ca^{2+} / Mg^{2+} composition of the culture medium (1.8 mM Ca^{2+} , 0.8 mM Mg^{2+}) in the Methods. In the revised figure legend of Figure 1A, we note that although EDTA chelates both cations, the inhibitory effect was consistent with that of Ca^{2+} -free medium and SOCE inhibition, supporting a Ca^{2+} -specific mechanism.

Comment 3-8

It would be helpful if the authors could provide estimates of the Ca^{2+} concentrations within the blebs in the discussion. However, this is not essential, as they clearly demonstrate a significant increase over basal Ca^{2+} in the cell body.

Response 3-8

We appreciate the reviewer's suggestion to provide estimates of the Ca^{2+} concentrations within expanding blebs. However, direct quantification of absolute Ca^{2+} concentrations in blebs is technically challenging, as the fluorescence intensity of genetically encoded Ca^{2+} indicators such as GCaMP is highly sensitive to environmental factors, including pH and viscosity, which differ between the bleb and the cell body. For this reason, we did not attempt to calculate absolute Ca^{2+} concentrations.

Nevertheless, we agree that obtaining quantitative measurements of Ca^{2+} concentration within blebs would be valuable for a more detailed understanding of Ca^{2+} dynamics in this process. Future studies combining ratiometric or FRET-based Ca^{2+} probes with in situ calibration methods, or the use of novel low-affinity Ca^{2+} sensors optimized for the cytoplasmic environment, may allow precise determination of local Ca^{2+} concentrations in blebs.

Comment 3-9

It should be specifically disclosed which form of GFP was used for fusion with CaMKII. I assume at least EGFP; and possibly even mEGFP with the further monomerizing mutation. Either way, this should be stated in the methods.

Response 3-9

We have now specified in the Methods that all fusion constructs used EGFP without additional monomerizing mutations.

Comment 3-10

In Fig. 6A, change "CaMKII" to "CaMKII DKO" in the image label.

Response 3-10

Corrected as requested. The figure and legend now indicate "CaMKII DKO."

Comment 3-11

It might be helpful to mention the cell type in every Fig legend (as three different cell types used overall).

Response 3-11

We have revised all figure legends to include the cell type used (DLD1 WT, CaMKII DKO, or Walker256), as appropriate.

Comment 3-12

The famous aminoacid T286 of CaMKII is correctly identified as T287 in the other isoforms. However, similarly, K42M in CaMKII alpha is K43M in the other isoforms. This should be corrected.

Response 3-12

We thank the reviewer for noting this. We have corrected the numbering throughout the manuscript: the ATP-binding-deficient mutant is referred to as **K43M** in CaMKII γ (corresponding to K42M in CaMKII α). This clarification has been added to the *Methods* and figure legends where relevant.

Dear Dr. Ikenouchi,

Thank you for submitting a revised version of your manuscript. I sincerely apologise for the protracted assessment of your revised manuscript due to delays in reviewer comment submission and the holiday period here in Germany. I have received input from two of the original reviewers, who now find that their concerns have been sufficiently addressed and now support publication of the manuscript. Therefore, there now remain only a few editorial points that need to be addressed before I can extend the official acceptance of the manuscript:

1. Please submit keywords for your manuscript.
2. According to the journal's style guidelines, please update the "Data availability" section to state "This study includes no data deposited in external repositories." Further information can be found at <https://link.springer.com/partners/embo-press/editorial-policies#Data%20availability%20statement>.
3. Please rename "Declaration of interests" section into "Disclosure and competing interests statement".
4. Please update references according to The EMBO Journal style - where there are more than 10 authors on a paper, the first 10 should be listed, followed by 'et al.'
5. Please remove movie legends from the manuscript text file and zip together with each movie file.
6. Please move figure legends after the "References" section.
7. Figure panels for Fig 5A-C, EV1 A-C and Fig EV2 A-C are not mentioned in the manuscript text; please add the corresponding callouts.
8. During our routine image integrity checks, we observed that the blot images within the Appendix file appear pixelated under analysis. This is often a result of converting original 16-bit TIFF files to RGB format for publication. While this is not inherently problematic, it can give the impression of image alteration to critical readers. To address this, please upload the Appendix file in a higher resolution. If it is not possible to reproduce the Appendix at a higher resolution, we recommend uploading the original blot and microscopy source data with your online submission instead.
9. In our standard source data check, we have noted unexplained numerical duplications in the source data for Fig. 2C. I have attached the corresponding files with the detected duplications labelled in colour - please note the two blocks of duplicated numerical values in the first column. Please check and correct as needed.
10. Our data editors have flagged the following issues in figure legends that need correcting:
 - Please provide the exact p values in the legends of figures 1B, 2C, 3C, E; 4B, 5C, F, H; 6F, I; 7B, D, E, G, I; EV3 B, EV4 C, F, G.
 - Please provide information on the number and nature of replicates in the legend of figure EV2 C.
 - Please define the error bars in the legends of figures EV1 C, EV2 C.
 - Please define the scale bar for figures 1A, EV3 A.
11. Papers published in The EMBO Journal are accompanied online by a 'Synopsis' to enhance discoverability of the manuscript. It consists of A) a short (1-2 sentences) summary of the findings and their significance, B) 3-4 bullet points highlighting key results and C) a synopsis image that is 550x300-600 pixels large (width x height, jpeg or png format). You can either show a model or key data in the synopsis image. Please note that the image size is rather small and that text needs to be readable at the final size.

With best wishes,

Ieva

Ieva Gailite, PhD
Senior Scientific Editor
The EMBO Journal
Meyershofstrasse 1
D-69117 Heidelberg
Tel: +4962218891309
i.gailite@embojournal.org

We realize that it is difficult to revise to a specific deadline. In the interest of protecting the conceptual advance provided by the work, we recommend a revision within 3 months (2nd Apr 2026). Please discuss the revision progress ahead of this time with the editor if you require more time to complete the revisions.

Referee #1:

The authors addressed my comments properly. As far as I am concerned, depending on the view of the other referees (who criticized other aspects of the paper), the paper can be accepted.

Referee #2:

The have addressed my concerns with the extensive revisions they have made to their manuscript. I think the major conclusions are well-supported and provides a new way of thinking about bleb dynamics in a large spectrum of cell contexts.

The authors addressed the remaining editorial issues.

Dear Junichi,

Thank you for incorporating the final editorial requests in your manuscript. I am now pleased to inform you that your manuscript has been accepted for publication. Congratulations with a nice study!

Before we forward your manuscript to the publishers, I will look into the synopsis text that you kindly provided and will let you know at the beginning of the next week if any edits to the journal style are needed.

Please note that it is The EMBO Journal policy for the transcript of the editorial process (containing referee reports and your response letters) to be published as an online supplement to each paper. If you should prefer removal of any referee-only figures included in the point-by-point response(s), e.g. because they may still be used for future publication or because they have been reproduced from published work by others, please do let us know immediately via response email.

More information is available here: <https://link.springer.com/partners/embo-press/editorial-policies#Peer%20review>

You may qualify for financial assistance for your publication charges - either via a Springer Nature fully open access agreement or an EMBO initiative. Check your eligibility: <https://link.springer.com/journal/44318/how-to-publish-with-us>

If you have any questions, please do not hesitate to contact the Editorial Office or me directly. Thank you for this very interesting contribution to The EMBO Journal!

Best wishes,

leva

leva Gailite, PhD
Senior Scientific Editor
The EMBO Journal
Meyerhofstrasse 1
D-69117 Heidelberg
Tel: +4962218891309
i.gailite@embojournal.org